# Future changes in Antarctic near-surface winds: regional variability and key drivers under a high-emission scenario

Cécile Davrinche<sup>1</sup>, Anaïs Orsi<sup>1,2</sup>, Charles Amory<sup>1,3</sup>, Christoph Kittel<sup>3,4,5</sup>, and Cécile Agosta<sup>1</sup>

Correspondence: Cécile Davrinche, davrinche.cecile@gmail.com

**Abstract.** Antarctic near-surface winds play a key role in shaping the local climate of Antarctica. For instance, they trigger drifting snow and reduce the amount of precipitation reaching the ground. Despite their importance, substantial uncertainties remain regarding their future changes over the continent associated with global warming, especially in winter. Here, we analyse projections of winter near-surface winds in Antarctica produced by four CMIP6 Global Climate Models downscaled by a regional atmospheric model adapted for the study of polar regions. Our analysis first demonstrates that the downscaling helps to improve the representation of near-surface winds at present day. On the continent, projected changes in July wind speeds between the late  $21^{st}$  and  $20^{th}$  centuries reveal considerable regional variability, with opposing trends depending on the area and model used. Nevertheless, the 4 models used agree on a significant strengthening of near-surface winds in Adélie Land, Ross ice shelf and Enderby Land and a significant weakening in some coastal areas, such as the Shackleton ice shelf, the Amundsen embayment region and the Filchner ice shelf. Using the momentum budget decomposition, we separate and quantify the contributions of different drivers to future changes in wind speed. These drivers include local forcings related to the net radiative cooling by the iced surface as well as large-scale forcing. We distinguish two types of local forcing: katabatic forcing (linked to the presence of a slope) and thermal wind forcing, which arises from horizontal gradients in the depth of the radiatively cooled surface layer. We project a significant decrease in both katabatic and thermal wind accelerations. Because in a warming climate they act to increase the wind speed in opposite directions, we find an overall compensation effect of the changes in katabatic and thermal wind at the margins of the continent, while large-scale forcing exhibits both significant increases and decreases depending on the location. Ultimately, we find that most significant strengthening of near-surface winds originate from strengthening in the large-sale forcing while most significant weakening of near-surface winds can be attributed to changes in the surface forcing.

<sup>&</sup>lt;sup>1</sup>Laboratoire de Sciences du Climat et de l'Environnement, LSCE-IPSL, CEA, CNRS, UVSQ, UMR8212, Université Paris Saclay, Gif-sur-Yvette, France

<sup>&</sup>lt;sup>2</sup>Department of Earth, Ocean and Atmospheric Sciences, The University of British Columbia, Vancouver, BC, Canada

<sup>&</sup>lt;sup>3</sup>Institut des Géosciences de l'Environnement (IGE), Université Grenoble Alpes/CNRS/IRD/G-INP, Grenoble, France

<sup>&</sup>lt;sup>4</sup>Laboratoire de Climatologie et Topoclimatologie, Département de Géographie, UR SPHERES, ULiège, Liège, Belgium

<sup>&</sup>lt;sup>5</sup>Physical geography research group, Department geography, Vrije Universiteit Brussel, Brussels, Belgium

#### 20 1 Introduction

40

The extraordinarily strong and persistent winds are a defining characteristic of Antarctica's climate. They include powerful westerlies over the ocean and easterlies at the ice sheet margins. In the interior, near-surface winds are predominantly directed downslope and play a major role in shaping the Antarctic climate as they trigger drifting snow (Amory, 2020), they indirectly influence sea ice formation (Holland and Kwok, 2012), the amount of precipitation reaching the ground (Grazioli et al., 2017), the stability of the boundary layer (Vignon et al., 2017) and they can play a determining role in triggering rapid ice shelf collapse (Cape et al., 2015).

Near-surface Antarctic winds result from both large-scale and surface pressure gradients (Van den Broeke and van Lipzig, 2002; Bintanja et al., 2014a; Davrinche et al., 2024), whose relative magnitudes in future projections are yet uncertain. Large-scale forcing is intrinsically linked to the leading modes of variability in the Southern Hemisphere: the Southern Annular Mode (SAM) and the El Niño–Southern Oscillation (ENSO). The SAM is quantified by the SAM index, which represents the zonally averaged sea-level pressure gradient between 40°S and 65°S (Marshall, 2003). ENSO is characterized by the Southern Oscillation Index (SOI), computed as the sea-level pressure difference between Tahiti and Darwin (Bromwich et al., 2004). Both SAM and ENSO influence the strength and position of the Amundsen Sea Low, a persistent low-pressure center in the Amundsen Sea sector (Raphael et al., 2016), which in turn modulates the frequency and trajectories of cyclones in West Antarctica (Fogt et al., 2012). In addition, surface forcing creates two additional pressure gradients. The first is a katabatic pressure gradient, which is proportional to the strength of the temperature inversion and the slope angle. This pressure gradient develops in sloped regions due to the quasi-permanent radiative cooling by the ice sheet (Phillpot and Zillman, 1970). The second is a local thermal wind pressure gradient, which is created by horizontal gradients in the depth of the temperature deficit layer. Thermal wind acts to replenish the pressure low created by the downslope displacement of air.

At present day, large-scale forcing dominates the variability of near-surface wind speed in the interior, while closer to the coast, both the katabatic and large-scale accelerations significantly contribute to the 3-hourly timescale variability (Davrinche et al., 2024). In future projections, however, the evolution of each family of forcing and their relative magnitude remains uncertain. On the one hand, the increase in greenhouse gases (GHG) concentration causes a decrease in net upward longwave radiation at the surface (Mitchell, 1989). As a consequence, the temperature inversion and thus the katabatic forcing should decrease (Van den Broeke and van Lipzig, 2002; Bintanja et al., 2014b). On the other hand, the increase in GHG concentration drives the SAM towards a more positive phase by the end of the 21<sup>st</sup> century (Miller et al., 2006; Fogt and Marshall, 2020; Goyal et al., 2021), while the effect on the SOI remains highly uncertain (Beobide-Arsuaga et al., 2021; Ren and Liu, 2025). Thus, models predict a strengthening and poleward shift of the westerlies, and a weakening of coastal off-shore easterlies during summer (Bracegirdle et al., 2008; Langlais et al., 2015; Hazel and Stewart, 2019; Neme et al., 2022). However, the trend of the large-scale forcing over the continent itself is unknown. In winter, changes in the zonally averaged SAM are indeed weaker. Therefore, Bracegirdle et al. (2008) hypothesized that the impact of the SAM does not have the ability to penetrate sufficiently southward to influence the large-scale forcing of coastal on-shore and mid-slope easterlies. However, under a doubling of CO<sub>2</sub>, Van Den Broeke et al. (1997) and Turner et al. (2013) showed that the circumpolar trough is locally

enhanced in specific locations where sea ice is completely removed (e.g., north of Ross and Amery ice shelves and north of the Peninsula). Although there is a consensus on the reduction of surface forcing in climate projections (van den Broeke et al., 2002; Bintanja et al., 2014b), large uncertainties remain regarding the evolution of the large-scale forcing around the coastlines of Antarctica in winter, and even more in the interior. Because of the zonal asymmetries in the changes of sea-level pressure around Antarctica, we expect to find zonal asymmetries in the evolution of the on-shore large-scale forcing as well.

Most studies on the future evolution of near-surface winds in Antarctica across different models focus on direct monthly wind speed output of GCMs (Neme et al., 2022; Bracegirdle et al., 2008). Davrinche et al. (2024) showed the importance of boundary layer processes in accurately representing the surface wind. However, GCMs often do not include an appropriate representation of the physics of the Antarctic boundary layer: Smith and Polvani (2017) show evidences of misrepresentation of the west-east Antarctica differences in the near-surface temperature field while Cuxart et al. (2000) mentions that GCMs commonly fail to represent the stability of the boundary layer. Here, we alleviate this shortcoming of GCMs by dynamically downscaling GCM with the polar-oriented regional atmospheric model MAR (Section 2.2). This ensures a better resolution of the ice sheet topography as well as a more realistic simulation of boundary layer dynamics achieved through adapted parametrizations of the interactions between the snow/ice surface and the atmosphere, as well as higher resolution vertical spacing near the surface.

In this paper, we investigate the projected changes of Antarctic winter near-surface winds under a high-emission scenario, focusing on the respective response of katabatic and large-scale forcings. We focus on the Antarctic continent, where slopes allow katabatic winds to form, and on the winter season, since it is the season for which both the katabatic forcing and the mean wind speed are the highest (Davrinche et al., 2024). We mitigate GCM limitations used in previous studies by using the regional atmospheric climate model MAR to dynamically downscale four recent CMIP6 GCMs carefully selected on their ability to represent the large-scale circulation in polar regions. We use the momentum budget decomposition to analyse how each family of drivers evolves in the different downscaled GCMs. In addition to Bintanja et al. (2014b), we evaluate the representativeness of the results by performing this analysis on four recent CMIP6 GCMs carefully selected on their ability to represent the large-scale circulation in polar regions. It enables us to mitigate single-model analysis issues and to test how robust potential changes are.

#### 2 Materials and Methods

80

#### 2.1 Selection of AWS using ERA5

#### 2.1.1 The AntAWS dataset

We use the monthly AntAWS dataset provided by Wang et al. (2023) that compiles all the available Automatic Weather Station (AWS) data in Antarctica from 1980 to 2021. For all 267 stations (except Zhongshan which is on a mast at  $\sim$ 10 m from the ground), data are collected at a height of  $\sim$ 3 m above ground level (agl), although the height of the wind sensor is poorly controlled and varies greatly between 1 and 6 m (Wang et al., 2023), depending on the initial sensor height and

snow accumulation rate. According to the logarithmic theoretical profile of wind speed in the boundary layer, with a constant roughness length  $z_0 = 1 \text{ mm}$  (Vignon et al., 2017), we estimate the maximum correction between wind speed measured at the real height of the sensor and wind speed at 3m to be between -10 % (for the correction from 1 to 3m) and 7 % (for the correction from 6 to 3m) of the theoretical value:

$$correction_{6-3} = \frac{log(\frac{6}{z_0})}{log(\frac{3}{z_0})} = 1.07$$
 (1)

$$correction_{1-3} = \frac{log(\frac{1}{z_0})}{log(\frac{3}{z_0})} = 0.90$$
 (2)

Data are collected every 3 hours and monthly averages are computed when at least 75 % of the 3-hourly observations are available in a month, based on Kittel et al. (2021). An additional quality control is performed in which wind speed exceeding  $60 \,\mathrm{m\,s^{-1}}$  or equal to  $0 \,\mathrm{m\,s^{-1}}$  are discarded. If wind speed and direction remain constant for 2 consecutive timesteps, values are discarded, as it might be due to sensors being frozen. Other values were flagged and validated or discarded based on a visual comparison with reanalysis datasets (ERA5). This includes rapidly changing values of wind speed (i.e., two consecutive values with a difference greater than 21  $\,\mathrm{m\,s^{-1}}$ ) and values outside of the likelihood interval of 3 standard deviations from the mean value, based on the criteria described in Lazzara et al. (2012).

## 2.1.2 ERA5 reanalysis

110

115

ERA5 is the latest reanalysis produced by the European Centre for Medium-Range Weather Forecasts (Hersbach et al., 2020). Its horizontal spatial resolution is ∼31 km and outputs are given at a hourly frequency. The assimilation system (IFS Cycle 41r2 4D-Var) uses 10 members to produce a 4D-Var ensemble of data assimilation (Hennermann and Guillory, 2019). Among various reanalysis products (MERRA-2, JRA-55, ERAI, NCEP2, and CFSR), ERA5 has been shown to perform best in capturing monthly averaged wind speeds (Dong et al., 2020).

## 105 2.1.3 Selection of AWS based on dataset length, and computation of the reference climatology

We want to create a climatology of the winter wind speed in Antarctica in order to have a reference to study the potential evolution of wind speed by the end of the  $21^{st}$  century. Therefore, we need datasets long enough to accurately represent the historical climatology. AWS data are only available during austral winters for almost 50 % (128 out of 267) of the stations. For computational cost purposes, our study focuses on the winter month of July. We screen for the availability of observations during this month. In order to test whether datasets are long enough to be representative of a climatological period, we compute using ERA5 the minimum value of  $N_{July}$  for which the standard error on the mean value of the July wind speed between 1980 and 2020 is inferior to 5 % of the mean value (see Supplementary Section S1.1). We conclude that selecting stations for which the number of July observations at each station  $N_{July}$  is greater than 10 is a reasonable criterion that enables a fair representation of the climatology of July wind speeds (Figure S1). As a result, out of 267 stations listed in the AntAWS dataset, we consider that only 28 of them are suitable to evaluate GCMs. These stations are presented in Fig. 1 and their elevation ranges from 30 to 3350 m above sea level (Table 1). For the 28 pre-selected AWS stations, the datasets exhibit no significant trend

**Figure 1.** Elevation, from Bedmachine 3 (Morlighem et al., 2020) (a) over all Antarctica, (b) zoomed on the black rectangle area. Superimposed are the 28 pre-selected AWS. Stations that have been discarded because of the inability of ERA5 to properly represent winds at these locations (see Sect. 2.1.4) are underlined. Red dashed contours indicate the Transantarctic mountains.

between 1980 and 2020, with values of the linear trend computed with ERA5 monthly July wind speed ranging between -0.08 and  $0.1 \text{ m s}^{-1} \text{ decade}^{-1}$ .

Furthermore, we compare the averaging of ERA5 wind speed over the 1980-2020 period or over the period available for each AWS, and we find differences lower than 0.4 m s<sup>-1</sup> in absolute value or 5 % of the mean value over 40 years (Figure S2). Therefore, we are confident that we can use the climatology at the 28 selected stations of the AntAWS dataset to evaluate the climatological historical mean of the GCMs over the period 1980-2000.

#### 2.1.4 Exclusion of sites near complex topography

GCMs have limited capacity to resolve local processes that influence regional climate, such as complex topography, land—sea contrasts and boundary layer convective processes (Di Virgilio et al., 2022). For a fair evaluation of GCMs, we do not want to analyze locations for which the topography is too specific and the resulting atmospheric dynamics will not be resolved by the models, e.g., close to the Transantarctic mountains or at the boundary between the ocean and the continent. We decided to exclude stations for which ERA5 wind speed in the nearest grid cell shows poor agreement with observed wind speed, as we do not expect GCMs to perform better than the reanalysis over the period of available AWS observations. We consider the following metrics, computed for monthly or annual means:

- the Pearson correlation coefficient (R) of ERA5 and AWS mean wind speed
- the normalized bias  $B = (|\overline{V_{ERA5}}| |\overline{V_{AntAWS}}|)/|\overline{V_{AntAWS}}|$
- and the normalized standard deviation  $\sigma_N = \sigma_{ERA5}/\sigma_{AntAWS}$

We compute these three metrics for July and for each station and we assign a score equal to 1 if |R| > 0.5 or  $B \le 30\%$  or  $0.5 < \sigma_N < 1.5$ , and -1 otherwise. Finally, we combine the scores into one total performance score (TPS) per station, computed as the sum of each individual performance score. This TPS is comprised between -3 and 3. Results are presented in Table 1 and Figure S3. We discard stations with a negative TPS, as it corresponds to half of the metrics exhibiting a poor performance score (Cape Bird, Windless Bight, Gill and Marble Point). These four stations exhibit the largest biases in terms of temporal variability (R < 0.3 and  $\sigma_N > 2$ , which indicates that the variability in ERA5 is underestimated) and mean amplitude (B > 30%, which indicates that ERA5 overestimates the mean value of the wind speed). Additionally, these stations are all located at the foot of the Transantarctic mountains (Fig. 1), which justifies their exclusion in the quantitative analysis. The 24 remaining stations, which cover locations from the coast to the plateau) are then listed in Table 1, above the double horizontal line.

#### 2.2 Climate models

150

155

#### 145 2.2.1 The regional atmospheric model MAR

The Regional Atmospheric Model MAR is a polar-oriented model which includes snowpack physics and its interactions with the atmosphere. It is a hydrostatic model whose primitive and prognostic equations have been extensively described in Gallée and Schayes (1994) and Gallée (1995). The turbulent scheme is well adapted to stable boundary layers, which is well suited for the study of polar regions. Additionally, the roughness length is parameterized as a function of surface air temperature to take into account the effect of sastrugis and is fitted to match observations of the temporal variability of wind speed in Adélie Land (Amory et al., 2017; Vignon et al., 2017; Agosta et al., 2019). The topography of the model is fixed, and derived from Bedmap 2 (Fretwell et al., 2013). We use 3-hourly model outputs on the standard Antarctic polar stereographic grid at a horizontal resolution of 35 km. The vertical spacing is in  $\sigma$  coordinates with 12 levels between  $\sim$ 2 m and  $\sim$ 1000 m above ground level. MAR is forced every 6 hours at the top of the atmosphere (wind and temperature, above 10 km) and at its lateral boundaries by large-scale atmospheric fields (wind, temperature, specific humidity, pressure, sea surface temperature, and sea ice concentration).

## 2.2.2 Selection of four Global Climate Models among CMIP6

We forced MAR with four GCMs from CMIP6: IPSL-CM6A-LR (Boucher et al., 2020), UKESM1-0-LL (Sellar et al., 2019), MPI-ESM1-2-HR (Mauritsen et al., 2019) and CNRM-CM6-1 (Voldoire et al., 2019). CMIP6 models are the latest GCM simulations from the Coupled Model Intercomparison Project (Eyring et al., 2016). Output of these models are regridded to MAR's 35km polar stereographic grid using a bilinear interpolation.

**Table 1.** List of AWS used to evaluate July wind speed and associated characteristics: longitude (Lon), latitude (Lat), elevation in MAR, real elevation, local slope in MAR and Total Performance Score (TPS, as described above). In the station name column, bracketed (C) corresponds to location where the corresponding grid-point of the model is at the interface between the continent and the ocean and bracketed (TM) correspond to locations close to the Transantarctic Mountains. The stations below the double horizontal line were excluded from the analysis, based on their low Total Performance Score (TPS, see Sec. 2.1.4)

| Station name           | $N_{July}$ | $\sigma/ \overline{\mathbf{V}} $ | Lon     | Lat    | Elevation | Real elevation | Slope                 | TPS |
|------------------------|------------|----------------------------------|---------|--------|-----------|----------------|-----------------------|-----|
|                        |            | (%)                              | (°)     | (°)    | (m, MAR)  | (m)            | $(\mathrm{mkm}^{-1})$ |     |
| D-47                   | 14         | 7.6                              | 138.73  | -67.39 | 1630      | 1560           | 7                     | 3   |
| D-10 (C)               | 14         | 6.2                              | 139.84  | -66.71 | 320       | 240            | 15                    | 3   |
| Clean Air              | 17         | 15.6                             | 0.0     | -90.0  | 2800      | 2840           | 2                     | 3   |
| Byrd                   | 15         | 11.8                             | -119.44 | -80.01 | 1520      | 1540           | 2                     | 3   |
| Elaine                 | 12         | 22.1                             | 174.24  | -83.07 | 70        | 60             | 1                     | 3   |
| Mizuho                 | 14         | 9.7                              | 44.29   | -70.7  | 2280      | 2260           | 4                     | 3   |
| Schwerdtfeger (TM)     | 32         | 19.7                             | 170.36  | -79.82 | 60        | 50             | 0                     | 3   |
| Relay Station          | 20         | 10.9                             | 43.06   | -74.02 | 3350      | 3350           | 2                     | 3   |
| Laurie II (C, TM)      | 13         | 17.8                             | 170.74  | -77.43 | 0         | 30             | 0                     | 3   |
| Henry                  | 18         | 12.8                             | -0.41   | -89.0  | 2830      | 2880           | 1                     | 3   |
| Ferrell (C, TM)        | 14         | 18.1                             | 170.82  | -77.78 | 40        | 40             | 4                     | 3   |
| Erin                   | 13         | 8.3                              | -128.87 | -84.9  | 920       | 990            | 6                     | 3   |
| Theresa                | 20         | 13.5                             | -115.85 | -84.6  | 1740      | 1450           | 10                    | 1   |
| Dome C                 | 13         | 16.5                             | 123.0   | -74.5  | 3230      | 3280           | 1                     | 1   |
| Dome C II              | 23         | 18.4                             | 123.35  | -75.11 | 3260      | 3250           | 0                     | 1   |
| Baldrick               | 12         | 6.7                              | -13.05  | -82.77 | 1970      | 1970           | 3                     | 1   |
| aws05                  | 14         | 11.8                             | -13.17  | -73.1  | 450       | 360            | 8                     | 1   |
| aws06                  | 11         | 10.4                             | -11.52  | -74.47 | 1050      | 1160           | 9                     | 1   |
| aws09                  | 20         | 16.7                             | 0.0     | -75.0  | 2870      | 2900           | 1                     | 1   |
| Marilyn (TM)           | 21         | 17.8                             | 165.77  | -79.9  | 60        | 60             | 0                     | 1   |
| Willie Field (TM)      | 16         | 13.9                             | 166.92  | -77.87 | 20        | 10             | 3                     | 1   |
| Vito (C)               | 11         | 15.0                             | 177.83  | -78.41 | 50        | 50             | 0                     | 1   |
| Lettau                 | 21         | 19.4                             | -174.59 | -82.48 | 60        | 40             | 0                     | 1   |
| Nico                   | 20         | 15.0                             | 90.02   | -89.0  | 3020      | 2980           | 2                     | 1   |
| Marble Point (C, TM)   | 34         | 15.1                             | 163.75  | -77.44 | 70        | 110            | 10                    | -1  |
| Gill                   | 12         | 15.1                             | -178.54 | -79.82 | 50        | 50             | 0                     | -1  |
| Windless Bight (C, TM) | 15         | 15.0                             | 167.67  | -77.73 | 30        | 40             | 6                     | -3  |
| Cape Bird (C, TM)      | 16         | 18.5                             | 166.44  | -77.22 | 0         | 40             | 5                     | -3  |

**Table 2.** List of selected GCMs with climate characteristics: Earth's equilibrium Climate Sensitivity (ECS) (Flynn and Mauritsen, 2020), horizontal resolutions (Williams et al., 2024), and storyline of projected Sea Ice Extent (SIE) and Stratospheric Polar Vortex (SPV) strength (Williams et al., 2024). SIE + (SIE-) corresponds to a storyline with a low (strong) projected SIE (when compared to the multi-model mean of CMIP6) while SPV+ (SPV-) corresponds to a storyline with a strong (weak) projected SPV strength.

| Model         | Institution | Resolution | ECS  | Winter | storyline |
|---------------|-------------|------------|------|--------|-----------|
|               |             |            |      | SIE    | SPV       |
| IPSL-CM6A-LR  | IPSL        | 250 km     | 4.50 | +      | +         |
| UKESM1-0-LL,  | MOHC        | 250 km     | 5.31 | +      | -         |
| MPI-ESM1-2-HR | MPI-M       | 100 km     | 2.84 | -      | +         |
| CNRM-CM6-1    | CNRM-       | 250 km     | 4.81 | -      | +         |
|               | CERFACS     |            |      |        |           |

CMIP6 models are selected based on their ability to represent the current climate at both poles ( $>50^{\circ}$  N in the Arctic and  $<40^{\circ}$ S in Antarctic). For this selection, nine metrics are considered: annual 500 hPa geopotential height, annual sea level pressure, summer sea surface temperature, winter sea ice concentration, annual and summer temperatures at 850 and 700 hPa.

165

We chose to study CMIP6 GCMs that are representative of a large range of climate sensitivity typical of CMIP6, and have a low fraction of implausibility for both poles and for all metrics. "Fraction of implausibility" is defined for each metric as the portion of the surface where the difference between historical averages in the model and ERA5 is greater than a plausible threshold set at 3 times the ERA5 interannual standard deviation (Agosta et al., 2022). This leads us to select IPSL-CM6A-LR, UKESM1-0-LL, MPI-ESM1-2-HR and CNRM-CM6-1 referred to in this paper as IPSL, UKESM, MPI and CNRM. Note that all of these models are Earth System Models, except for CNRM-CM6-1 which does not include interactive ocean biogeochemistry nor atmospheric chemistry (Voldoire et al., 2019).

The choice of these four models for our study is supported by another study by Williams et al. (2024) where these models were classified among the best performing in winter when comparing their sea ice extent (SIE), surface air temperature, zonal wind at 850 and 50 hPa to ERA5. Furthermore, these models are representative of the large variability of plausible patterns of responses to climate change among CMIP6 models and can be expected to exhibit different patterns in wind-speed changes by the end of the 21<sup>st</sup> century. For example, Williams et al. (2024) noted that they correspond to different storylines for Antarctica, using winter SIE and Stratospheric Polar Vortex (SPV, linked to the strength and position of the surface westerlies, Table 2) as predictors. Additionally, they have different Earth's Equilibrium Climate Sensitivity (ECS, corresponding to the change in temperature at equilibrium that would result from a doubling of CO<sub>2</sub>), which is a proxy for the intensity with which the model warms the Earth's surface temperature. While UKESM has one of the strongest ECS of all CMIP6 models, MPI exhibits one of the lowest.

# 2.2.3 Experiments

We use a high emission scenario (SSP585) to test the sensitivity of wind speed to climate change with a strong warming of the continent. The expected global radiative forcing by 2100 with this scenario is +8.5 W m<sup>-2</sup> (IPCC AR6, 2023). We then force MAR by one member of each of the four GCMs (r1i1p1f1 for all models except CNRM-CM6A-1, which is forced by r1p1i1f2). Here, we define the historical reference period as 1980-2000 and compare this period with the end of the 21<sup>st</sup> century (2080-2100), as in Bracegirdle et al. (2020). We study the change in the monthly-mean July near-surface wind speed at 10 m (sfcWind in CMIP6) averaged over 20 years, between these two periods.

# 2.2.4 Statistical significance

In order to test the statistical significance of changes in 10 m wind speed or any related variable between the end of the 21<sup>st</sup> and the 20<sup>th</sup> century, we apply the non-parametric Kruskal-Wallis test (Kruskal and Wallis, 1952). This test (also called one-way ANOVA on rank) is performed at a level of significance of 80 %. It has been used in multiple previous studies to assess past or future changes (Machado and Calliari, 2016; Marshall et al., 2017; Casado et al., 2023, e.g.,). This test assesses that one sample (e.g., July mean monthly wind speed between 2080 and 2100) has significantly higher or lower values than another one (e.g., July mean monthly wind speed between 1980 and 2000).

#### 2.3 Momentum budget decomposition

#### 2.3.1 Equations

The momentum budget decomposition is a useful tool for identifying the drivers of wind speed variability in Antarctica (Van den Broeke and van Lipzig, 2002; Bintanja et al., 2014b). The method is described extensively in Davrinche et al. (2024). For each model downscaled by MAR, we compute the momentum budget in the cross- and downslope directions and we decompose it into 6 different accelerations, defined as follows:

|     |                                   | Horizontal                                                         | Coriolis | Vertical advection                                                       | Large-scale    | Thermal wind                                                  | Katabatic |
|-----|-----------------------------------|--------------------------------------------------------------------|----------|--------------------------------------------------------------------------|----------------|---------------------------------------------------------------|-----------|
|     |                                   | advection                                                          |          | & Turbulence                                                             |                |                                                               |           |
|     | Cross-slope:                      | ADVH                                                               | COR      | TURB                                                                     | $\mathbf{LSC}$ | $\mathrm{THW}_{\mathrm{TD}}$                                  | KAT       |
| 205 | $\frac{\partial U}{\partial t} =$ | $-U\frac{\partial U}{\partial x} - V\frac{\partial U}{\partial y}$ | +fV      | $-W\frac{\partial U}{\partial z} - \frac{\partial u\bar{w}}{\partial z}$ | $-fV_{LSC}$    | $+\frac{g}{\theta_0}\frac{\partial \hat{\theta}}{\partial x}$ |           |
|     | Downslope:                        |                                                                    |          |                                                                          |                |                                                               |           |
|     | 0.7.7                             | 017 017                                                            |          | 017 0 -                                                                  |                | o ô                                                           |           |

 $\frac{\partial V}{\partial t} = -U\frac{\partial V}{\partial x} - V\frac{\partial V}{\partial y} \qquad -fU \qquad -W\frac{\partial V}{\partial z} - \frac{\partial v\overline{w}}{\partial z} \qquad +fU_{LSC} \qquad +\frac{g}{\theta_0}\frac{\partial \hat{\theta}}{\partial y} \qquad +\frac{g}{\theta_0}\Delta_{\theta}\sin(\alpha) \qquad (3)$ where (U,V) are the horizontal components of the wind in the cross, and desirable actions  $\phi$  is the local class  $\theta$  is the

where (U, V) are the horizontal components of the wind in the cross- and downslope direction,  $\alpha$  is the local slope,  $\theta$  is the potential temperature, and  $\theta_0$  is the background potential temperature described in Davrinche et al. (2024).  $\theta_0$  represents the

extrapolation down to the surface of the potential temperature in the upper part of the atmosphere, where surface processes do not come at play.  $\Delta\theta$  represents the temperature deficit, i.e., the difference between the background and the actual potential temperature.  $\hat{\theta}$  is the vertically integrated potential temperature deficit from the top of the inversion layer. Above the inversion layer, as  $\theta = \theta_0$ , both  $\Delta\theta$  and  $\hat{\theta}$  become zero. While the latter are linked to the influence of the surface on the vertical potential temperature profile,  $\theta_0$  is related to the synoptic forcing and is used in the computation of the large-scale components of the winds  $V_{LSC}$  and  $U_{LSC}$ :

$$\begin{cases}
\frac{\partial U_{LSC}}{\partial \ln(p)} = +\frac{R_d}{f} \left(\frac{p}{p_0}\right)^{\frac{R_d}{Cp}} \left(\frac{\partial \theta_0}{\partial y}\right)_p \\
\frac{\partial V_{LSC}}{\partial \ln(p)} = -\frac{R_d}{f} \left(\frac{p}{p_0}\right)^{\frac{R_d}{Cp}} \left(\frac{\partial \theta_0}{\partial x}\right)_p
\end{cases}$$
(4)

where p is the pressure (in hPa),  $p_0$  the standard reference pressure (equals to 1013.2 hPa),  $R_d$  and  $C_p$  are respectively the gas constant and specific heat capacity of dry air ( $R_d$  = 287 J kg<sup>-1</sup> K<sup>-1</sup> and  $C_p$ = 1005.7 J kg<sup>-1</sup> K<sup>-1</sup>).

Further descriptions of the equations and validation of the method is performed in Davrinche et al. (2024).

#### 2.3.2 Description of the six accelerations

The pressure gradient force (PGF) in the momentum budget equation is divided into three accelerations reflecting the origin of the driver: the large-scale acceleration, katabatic acceleration, and the thermal wind acceleration. The large-scale acceleration (**LSC**) represents the portion of the PGF that originates from the synoptic forcing above the boundary layer. The katabatic acceleration (**KAT**) represents the gravity-driven motion induced by the temperature inversion over a sloping surface. It is especially strong in austral winter in a narrow band close to the coastal margins. It exhibits a strong diurnal cycle in summer and seasonal cycle throughout the year. The thermal wind acceleration (**THW**<sub>TD</sub>), related to the temperature deficit, is sometimes referred to as shallow baroclinicity (Caton Harrison et al., 2024) or integrated temperature deficit (Parish and Cassano, 2003). It corresponds to the near-surface baroclinicity induced by changes in the depth of the temperature deficit layer. In the rest of the study, special attention will be given to these PGF-related accelerations. They are indeed considered as active terms (Van den Broeke and van Lipzig, 2002) as they are produced by a forcing, either large-scale or surface pressure gradients.

In addition, three other passive accelerations contribute to the momentum budget. They form as a reaction to an existing motion that has been triggered by an active term. First, there is the horizontal advection (**ADVH**), which corresponds to the horizontal transport of momentum budget by the wind itself. It is weak in comparison to the other terms of the momentum budget equations but can sometimes become significant in coastal areas or in topographically complex zones such as valleys, or at the foot of the mountains. Then, there is the Coriolis acceleration (**COR**). It is a deviation induced by the Earth's rotation and it results in a rotation of the wind by 90° to the west in comparison to its acceleration. Lastly, the residual term (**TURB**) encompasses vertical advection (which is weak), turbulent drag (which opposes the other accelerations and is strong when the wind speed is high) and potential errors arising from closing the momentum budget. A comparison of MAR's native turbulent acceleration and our recomputed residual turbulence as detailed in Davrinche et al. (2024) enables us to conclude that the error resulting from closing the budget in July is small compared to the absolute value of the turbulence (i.e., ~10 % for all models).

# 2.3.3 Attribution of changes in wind speed using the Momentum Budget Decomposition

In winter, the first order temporal derivatives of the wind vector  $(\frac{\partial U}{\partial t})$  and  $\frac{\partial V}{\partial t}$  are 5 orders of magnitude smaller than the other accelerations (Fig. 2). Therefore, we can assume stationary conditions and rewrite Eq. (3) in a "quasi-geostrophic" form:

$$\begin{cases}
U = \underbrace{\frac{1}{f} \left( -U \frac{\partial V}{\partial x} - V \frac{\partial V}{\partial y} \right)}_{U_{ADVH}} + \underbrace{\frac{1}{f} \left( -W \frac{\partial V}{\partial z} - \frac{\partial v \bar{w}}{\partial z} \right)}_{U_{TURB}} + \underbrace{\frac{1}{f} \left( f U_{LSC} \right)}_{U_{LSC}} + \underbrace{\frac{1}{f} \left( \frac{g}{\theta_0} \frac{\partial \hat{\theta}}{\partial x} \right)}_{U_{THW}} + \underbrace{\frac{g}{f\theta_0} \Delta \theta \sin(\alpha)}_{U_{KAT}} \\
V = \underbrace{-\frac{1}{f} \left( -U \frac{\partial U}{\partial x} - V \frac{\partial U}{\partial y} \right)}_{V_{ADVH}} - \underbrace{\frac{1}{f} \left( -W \frac{\partial U}{\partial z} - \frac{\partial u \bar{w}}{\partial z} \right)}_{V_{LSC}} + \underbrace{\frac{1}{f} \left( f V_{LSC} \right)}_{V_{THW}} - \underbrace{\frac{1}{f} \left( \frac{g}{\theta_0} \frac{\partial \hat{\theta}}{\partial y} \right)}_{V_{THW}} 
\end{cases} (5)$$

The vectorial form of this equation is:

$$V = V_{ADVH} + V_{TURB} + V_{LSC} + V_{THW} + V_{KAT}, \tag{6}$$

with V the total wind vector, of components (U, V) in the cross- and downslope coordinate system, and  $V_{ACC}$  the wind that would be in geostrophic balance with the corresponding acceleration ACC (i.e., Coriolis acceleration balances ACC), of components  $(U_{ACC}, V_{ACC})$  shown in Eq. (5). Note that the wind vector associated to each acceleration corresponds to a rotation to the left of the acceleration, with the norm divided by 1/f. For example, the **KAT** acceleration is downslope, but its contribution to the wind vector  $V_{KAT}$  is in the cross-slope direction due to its deviation by Coriolis.

We define |V| as the norm of the wind vector (i.e., the wind speed). This norm can be written as the scalar product of the wind direction  $\frac{V}{|V|}$  with the wind vector, which enables us to decompose the wind speed into a sum of contributions:

$$|\mathbf{V}| = \frac{\mathbf{V}}{|\mathbf{V}|} \cdot \mathbf{V} \tag{7}$$

$$255 \implies |\mathbf{V}| = \frac{\mathbf{V}}{|\mathbf{V}|} \cdot \mathbf{V_{ADVH}} + \frac{\mathbf{V}}{|\mathbf{V}|} \cdot \mathbf{V_{TURB}} + \frac{\mathbf{V}}{|\mathbf{V}|} \cdot \mathbf{V_{LSC}} + \frac{\mathbf{V}}{|\mathbf{V}|} \cdot \mathbf{V_{THW}} + \frac{\mathbf{V}}{|\mathbf{V}|} \cdot \mathbf{V_{KAT}}. \tag{8}$$

Projected changes in near-surface wind speed between the end of the  $21^{st}$  and the end of the  $20^{th}$  century  $\Delta |\overline{\mathbf{V}}|$  can be decomposed as the sum of changes in the mean value of the scalar product computed on 3-hourly values of each accelerations with the wind direction vector:

$$\Delta \overline{|\mathbf{V}|} = \Delta \overline{\frac{\mathbf{V}}{|\mathbf{V}|} \cdot \mathbf{V_{ADVH}}} + \Delta \overline{\frac{\mathbf{V}}{|\mathbf{V}|} \cdot \mathbf{V_{TURB}}} + \Delta \overline{\frac{\mathbf{V}}{|\mathbf{V}|} \cdot \mathbf{V_{LSC}}} + \Delta \overline{\frac{\mathbf{V}}{|\mathbf{V}|} \cdot \mathbf{V_{THW}}} + \Delta \overline{\frac{\mathbf{V}}{|\mathbf{V}|} \cdot \mathbf{V_{KAT}}}$$
(9)

Therefore, changes in near-surface wind speed between the end of the  $21^{st}$  and the end of the  $20^{th}$  century can be decomposed as a sum of scalar product (Fig. 2). In the rest of the paper, we will note  $\Delta ACC$  the "changes in wind speed due to a specific acceleration between 2080-2100 and 1980-2000", with ACC being the specific term considered (LSC, THW, KAT, ADVH, TURB), that we define as follows:

$$\Delta ACC = \frac{\overline{\mathbf{V}} \cdot \mathbf{V_{ACC}}}{|\mathbf{V}|} \cdot \mathbf{V_{ACC}}(2080 - 2100) - \frac{\overline{\mathbf{V}} \cdot \mathbf{V_{ACC}}}{|\mathbf{V}|} \cdot \mathbf{V_{ACC}}(1980 - 2000). \tag{10}$$

Figure 2. (a) Changes in multi-model mean (MAR-IPSL, MAR-UKESM, MAR-MPI and MAR-CNRM) 10-m wind speed between 2080-2100 and 1980-2000, (b) Changes in multi-model mean (MAR-IPSL, MAR-UKESM, MAR-MPI and MAR-CNRM) 10-m scalar product of the sum of the accelerations with the wind direction  $\frac{\mathbf{V}}{|\mathbf{V}|}$ , i.e.,  $\frac{\mathbf{V}}{|\mathbf{V}|} \cdot \mathbf{V}_{\mathbf{SUM}} = \frac{\mathbf{V}}{|\mathbf{V}|} \cdot (\mathbf{V}_{\mathbf{ADVH}} + \mathbf{V}_{\mathbf{TURB}} + \mathbf{V}_{\mathbf{LSC}} + \mathbf{V}_{\mathbf{THW}} + \mathbf{V}_{\mathbf{KAT}})$ , between 2080-2100 and 1980-2000 and (c) Difference of (a) and (b)

Table 3. Improvement of the mean bias due to downscaling of the 4 GCMs in July (first 3 columns) and annually (last 3 columns). The improvement of the mean bias is computed as the difference between the absolute values of mean normalised bias of the monthly wind speed output of GCMs (compared to AWS measurements) and the absolute values of mean normalised bias of the monthly wind speed output of GCMs downscaled by MAR ( $|B_{GCM}| - |B_{MAR-GCM}|$  in %). Positive values indicate an improvement due to downscaling while negative values indicate a decline. Significant improvements due to downscaling (computed using a t-test with a significance level of 0.1) are denoted by an asterisk (\*). Values are given for the 28 AWS for which there is enough July months to create a climatology, for the 24 AWS presented in Table 1 that exhibit a coherent representation of the wind in ERA5 and for the 18 stations listed in Table 1 that are not in the Transantarctic mountains, nor on the shore (without TM/C)

|                    |      | July  |         | Annually |         |         |  |  |
|--------------------|------|-------|---------|----------|---------|---------|--|--|
| Improvement due to | 28   | 24    | without | 28       | 24      | without |  |  |
| downscaling (%)    | AWS  | AWS   | (TM/C)  | AWS      | AWS     | (TM/C)  |  |  |
| IPSL               | +4.4 | +6.9* | +9.3*   | +9.0*    | +8.8*   | +11.7*  |  |  |
| UKESM              | +1.3 | +8.3  | +9.8*   | +7.0*    | +11.1*  | +12.1*  |  |  |
| MPI                | +0.2 | +5.9  | +10.7*  | +8.1*    | +10.0 * | +16.0*  |  |  |
| CNRM               | -0.2 | +1.6  | +3.1*   | +1.8*    | +1.6*   | +4.1*   |  |  |

#### 3 Results

# 3.1 Evaluation of the models ability to represent near-surface winds in Antarctica

We evaluate the value of the downscaling by comparing biases in monthly mean 10-m wind speed computed between weather station observations (see Sect. 2.1) and GCMs alone or downscaled by MAR (Fig. 3).

Figure 3. (a) Altitude of the selected stations. Mean normalized bias (B) for wind speed with regard to the AntAWS observations (B =  $(|\overline{\mathbf{V}_{\mathbf{GCM}}}| - |\overline{\mathbf{V}_{\mathbf{AntAWS}}}|)/|\overline{\mathbf{V}_{\mathbf{AntAWS}}}|$  for (a) and B =  $(|\overline{\mathbf{V}_{\mathbf{MAR-GCM}}}| - |\overline{\mathbf{V}_{\mathbf{AntAWS}}}|)/|\overline{\mathbf{V}_{\mathbf{AntAWS}}}|$  for (b)) for the 24 selected AntAWS stations, computed for July (b) using the GCMs, (c) using the GCMs downscaled by MAR.

Overall, all GCMs tend to underestimate the mean wind speed, with the mean normalised bias across the 24 stations ranging from -24 % for MPI, which demonstrates a consistent negative bias at all stations, to -13 % for CNRM (Fig. 3b). The latter exhibits indeed a slight positive bias in coastal locations (Willie Field, Gill, Vito and D-10) that is compensated for by a negative bias everywhere else (Fig. 3b). In contrast, UKESM shows an inverse pattern, displaying substantial negative biases in coastal stations that are partially offset by a pronounced positive bias at Dome C on the plateau.

We observe that biases are more similar between models downscaled by MAR than for raw GCMs (Fig. 3c), except for Dome C and Dome C II in MAR-CNRM. Furthermore, the downscaling by MAR significantly reduces the mean bias compared to the different GCMs in the sloped regions of Antarctica i.e., from AWS05 at 360 m above sea level (Fig. 3a) to Henry at 2880 m above sea level), where topography plays an important role in shaping the wind field. However, there is a consistent overestimation of the weak winds of the Plateau across all downscaled models and an underestimation of the stronger winds in coastal areas. Downscaling by MAR reduces the regional variability in wind speed bias on the continent.

Overall, downscaling by MAR significantly reduces the mean biases of the different GCMs, with the exception of stations situated at the interface between the continent and the ocean (i.e., D-10) or in the Transantarctic mountains (Willie Field, Lettau, Schwerdtfeger, Marilyn, Ferrell, and Lettau) (Fig. 3). With these coastal and Transantarctic AWS, there is a significant

improvement of the mean normalised bias for all models annually, but in July, improvements are not statistically significant (Table 3). However, if we discard the coastal and Transantarctic AWS, there is a significant improvement of the mean normalised bias for all models and in all seasons.

To conclude, downscaling with a regional climate model significantly improves the representation of near-surface winds. A finer resolution helps with topographic forcing, but the improved physics likely provides benefits in sloped terrains and on the plateau.

# 3.2 Projected changes in near-surface winds by the end of the $21^{st}$ century

In winter, all downscaled GCMs project a strengthening and poleward shift of the westerlies over the ocean (Fig. 4a and b), more pronounced in MAR-IPSL and MAR-UKESM, which are also the models with the strongest changes in sea ice concentration. On the continent, changes are weaker, with larger differences among the downscaled models. Each of them features approximately 50 % of the continental grid cells exhibiting an increase and 50 % exhibiting a decrease in wind speed by the end of the 21<sup>st</sup> century (Table 4). The ratio of significant decrease and significant increase remains approximately equal, both under 20 % except for MAR-IPSL which exhibits more significant increases (40 %) than significant decreases (6 %). Regions of significant changes greatly vary among the downscaled models, with more significant decrease in coastal areas for MAR-UKESM and MAR-CNRM, large patches of significant increases on the East Antarctic Plateau for MAR-IPSL and smaller-size sparse patches for MAR-MPI (Fig. 4).

**Table 4.** Percentage of continental grid cells (including ice shelves) exhibiting an increase in July wind speed between 2080-2100 and 1980-2000 (significant or not,  $\Delta |\overline{\mathbf{V}}| > 0$ ), a significant increase in wind speed ( $\Delta |\overline{\mathbf{V}}| > 0^*$ ), no significant change in wind speed ( $\Delta |\overline{\mathbf{V}}| \sim 0$ ), a significant decrease in wind speed ( $\Delta |\overline{\mathbf{V}}| < 0^*$ ) and a decrease in wind speed (significant or not,  $\Delta |\overline{\mathbf{V}}| < 0$ ), for MAR-IPSL, MAR-UKESM, MAR-MPI. MAR-CNRM, for at least 3 downscaled models (>3M) and for the multi-model mean (MAR-MMM)

| Model     | $\Delta  \overline{\mathbf{V}}  > 0$ | $\Delta  \overline{\mathbf{V}}  > 0*$ | $\Delta  \overline{\mathbf{V}}  \sim 0$ | $\Delta  \overline{\mathbf{V}}  < 0*$ | $\Delta  \overline{\mathbf{V}}  < 0$ |
|-----------|--------------------------------------|---------------------------------------|-----------------------------------------|---------------------------------------|--------------------------------------|
| MAR-IPSL  | 71 %                                 | 40 %                                  | 55 %                                    | 6 %                                   | 29 %                                 |
| MAR-UKESM | 42 %                                 | 11 %                                  | 76 %                                    | 13 %                                  | 58 %                                 |
| MAR-MPI   | 49 %                                 | 16 %                                  | 66 %                                    | 18 %                                  | 51 %                                 |
| MAR-CNRM  | 52 %                                 | 18 %                                  | 72 %                                    | 11 %                                  | 48 %                                 |
| >3M       | 41 %                                 | 8 %                                   | 90 %                                    | 2 %                                   | 35 %                                 |
| MAR-MMM   | 57 %                                 | 23 %                                  | 63 %                                    | 14 %                                  | 43 %                                 |

However, some areas display similar changes in all downscaled GCMs and in the multi-model mean (MAR-MMM, see right column in Fig. 4a). There is a significant increase on the Ross ice shelf (Fig. 4c(iii)) for all models except MAR-MPI, a significant increase on Enderby Land (Fig. 4c(vi)) for all models except MAR-UKESM, a significant increase in Adélie Land (Fig. 4c(ii)) for all models and a significant decrease for all models except MAR-IPSL on Shackleton ice shelf (Fig. 4c(i)), Filchner ice shelf (Fig. 4c(v)) and on in the Amundsen embayment region (Fig. 4c(iv)).

Figure 4. Projection of 10-m July wind speed changes between 2080-2100 and 1980-2000 ( $\Delta |\overline{\mathbf{V}|_{10m}}$ ) for GCMs downscaled by MAR (a) and for GCMs (b). MMM refers to the multi-model mean. Superimposed is the contour line at -30 % of the difference in Sea Ice Concentration (SIC) between July 2080-2100 and July 1980-2000 (black dashed line). (c) Map of the zones of significant near-surface wind speed changes between 2080-2100 and 1980-2000. Dark red (blue) areas represent zones for which at least 3 GCMs downscaled by MAR project a significant increase (decrease) of near-surface wind speed. Light red (blue) areas represent zones for which 2 models project a significant increase (decrease) of near-surface wind speed. Hashed grey areas indicate locations for which there is a significant disagreement between at least two models regarding the sign of evolution of near-surface wind speed. Green squares define 6 zones of interest which are used in the rest of the article: (i) Shackleton ice shelf, (ii) Adélie Land, (iii) Ross ice shelf, (iv) Amundsen embayment region, (v) Filchner ice shelf and (vi) Enderby Land.

**Table 5.** Percentage of the continental (including ice shelves) grid cells exhibiting an increase or a decrease in the scalar product of wind direction and large-scale wind (first three columns), katabatic wind (columns 4 to 6), thermal wind (column 7 to 9) and the sum of katabatic and thermal wind (column 10 to 12). Metrics are computed as differences of the average values over the months of July between 2080-2100 and 1980-2000 for different models. MMM indicates changes in the multi-model mean while >3M indicates significant changes observed in at least 3 downscaled GCMs.

|           | ΔLSC |      |          | $\Delta$ KAT |      | $\Delta \text{THW}$ |      |      | $\Delta { m SURF}$ |      |      |          |
|-----------|------|------|----------|--------------|------|---------------------|------|------|--------------------|------|------|----------|
| Model     | > 0* | < 0* | $\sim 0$ | > 0*         | < 0* | $\sim 0$            | > 0* | < 0* | $\sim 0$           | > 0* | < 0* | $\sim 0$ |
| MAR-IPSL  | 38 % | 2 %  | 60 %     | 3 %          | 36 % | 61 %                | 22 % | 11 % | 67 %               | 8 %  | 34 % | 58 %     |
| MAR-UKESM | 20 % | 4 %  | 76 %     | 4 %          | 37 % | 59 %                | 19 % | 12 % | 69 %               | 5 %  | 33 % | 62 %     |
| MAR-MPI   | 29 % | 2 %  | 69 %     | 5 %          | 50 % | 45 %                | 22 % | 13 % | 65 %               | 6 %  | 48 % | 46 %     |
| MAR-CNRM  | 25 % | 8 %  | 67 %     | 4 %          | 52 % | 44 %                | 29 % | 7 %  | 64 %               | 12 % | 43 % | 45 %     |
| >3M       | 9 %  | 0 %  | 91 %     | 1 %          | 33 % | 66 %                | 11 % | 2 %  | 87 %               | 3 %  | 27 % | 70 %     |
| MAR-MMM   | 48 % | 5 %  | 47 %     | 5 %          | 66 % | 29 %                | 34 % | 13 % | 53 %               | 10 % | 59 % | 31 %     |

Although downscaling by MAR significantly improves the representation of near-surface winds (Sec. 3.1), projected 10-m wind speed changes between 2080-2100 and 1980-2000 using GCMs not downscaled by MAR show similar patterns of evolution (e.g., an increase on the Ross ice shelf and in Adélie Land) but however miss out on most of the significant decreases in near-surface winds (compare Table 4 with Table S1).

# 3.3 Projected changes in the components of near-surface winds

#### 310 3.3.1 Changes in large-scale circulation

In every model, the increase in wind speed over the ocean is associated with an increase in the large-scale contribution (Fig. 5b), which is partially offset by an associated increase in turbulence (Fig. 5f). The Pearson correlation coefficient (R) between changes in wind speed over the ocean and changes in wind speed due to large-scale is greater than 0.7 for all models (Table S2). Note that MPI displays the weakest poleward shift and strengthening of the surface westerlies. It is also the model with the lowest ECS (Table 2), and the largest sea ice extent at present day.

This result is in agreement with previous studies that showed that the already observed increasing positive trend of the SAM is likely to continue in response to increasing GHG and after the recovery of the ozone hole (which offsets the strengthening of the SAM (Bracegirdle et al., 2008)). As a consequence of the increased pressure gradient between the mid-latitudes and 65 °S, westerlies are strengthening and shifting poleward (Goyal et al., 2021; Fyfe, 2006).

The pattern of increase in westerlies coincidentally appears to follow closely changes in the extent of sea ice, shown in thick black lines in Figure 4. For GCMs with low sea ice loss (IPSL and UKESM), the poleward shift of the westerlies does not extend up to the coastline in the Indian sector (20-90° E) in East Antarctica, while it does for models with strong sea ice extent

Figure 5. Projection of changes in 10-m wind speed between 2080-2100 and 1980-2000 associated with large-scale forcing  $(\Delta LSC = \frac{\overline{V}_{|V|} \cdot V_{LSC}}{|V|} \cdot V_{LSC}(2080 - 2100) - \frac{\overline{V}_{|V|} \cdot V_{LSC}}{|V|} \cdot V_{LSC}(1980 - 2000)$ , column b), katabatic forcing  $(\Delta KAT = \frac{\overline{V}_{|V|} \cdot V_{KAT}}{|V|} \cdot V_{KAT}(2080 - 2100) - \frac{\overline{V}_{|V|} \cdot V_{KAT}}{|V|} \cdot V_{KAT}(1980 - 2000)$ , column c), thermal wind forcing  $(\Delta THW = \frac{\overline{V}_{|V|} \cdot V_{THW}}{|V|} \cdot V_{THW}(2080 - 2100) - \frac{\overline{V}_{|V|} \cdot V_{THW}}{|V|} \cdot V_{THW}(1980 - 2000)$ , column d), advection  $(\Delta ADVH = \frac{\overline{V}_{|V|} \cdot V_{ADVH}}{|V|} \cdot V_{ADVH}(2080 - 2100) - \frac{\overline{V}_{|V|} \cdot V_{ADVH}}{|V|} \cdot V_{TURB}(2080 - 2100) - \frac{\overline{V}_{|V|} \cdot V_{TURB}}{|V|} \cdot V_{TURB}(1980 - 2000)$ , column f) and sum of all the above-mentioned forcings (large-scale, katabatic, thermal wind, advection and turbulence), which is equivalent to changes in wind speed  $(\Delta SUM = \Delta LSC + \Delta KAT + \Delta THW + \Delta ADVH + \Delta TURB$ , column a), see Fig. 2. Dotted areas indicate locations for which changes are significant at a 80 % level, significant area larger than 350 km² are highlighted with a grey solid line.

loss (MPI and CNRM). MPI retains a significant amount of sea ice in the Pacific sector at the end of the  $21^{st}$  century, where other models show a retreat, and thus does not show an increase in the large-scale wind as others do.

On the continent, the results are much less homogeneous. Most significant changes in large-scale acceleration are positive (48% in the MMM, Table 5) and some locations such as Adélie Land (Figure 7) or Enderby Land exhibit a significant increase in large-scale forcing in all models. Aside from these areas, models disagree on the exact location of significant changes: MAR-IPSL and MAR-UKESM project, for example, a significant strengthening of large-scale acceleration on the Ross ice shelf while MAR-MPI and MAR-CNRM projects a non-significant weakening. Everywhere else in Antarctica, MAR-IPSL and MAR-MPI project an overall increase in large-scale acceleration, while MAR-UKESM and MAR-CNRM exhibit some significant weakening of coastal easterlies (with minor changes in the mean wind direction) on Shackleton ice shelf and in Queen Maud Land (between Filchner ice shelf and Enderby Land). From Fig. 5b, we also observe that the largest inter-model differences in the forcing of wind changes originate from differences in the large-scale pattern of change.

These inconsistencies are related to variable trends in large-scale pressure gradients that are different between models. Although the trend in SAM is well understood and reproduced by most models (MPI does not show a clear trend), the changes in the pressure gradient between the circumpolar trough at 65 °S and the pole are much less clear and inconsistent between models. In Antarctica, computing the pressure gradient based on the mean sea level pressure results in strong biases because of the extrapolation of the pressure under the surface layer. Instead, we looked directly at the difference between the mean geopotential height and mean geopotential height at 65 °S at 500 hPa (Figure S4). For MAR-UKESM, on the interior, the difference with the geopotential height at 65 °S becomes more negative at the end of the 21st, meaning that the polar cell is strengthening. It is the opposite for MAR-IPSL and MAR-MPI, and there is on average no change for MAR-CNRM. However, we found no evidence of a correlation between a strengthening of the polar cell and an intensification of the large-scale pressure gradients at the surface. The attribution and robustness of changes in large-scale pressure gradients remain to be evaluated.

#### 3.3.2 Changes in surface forcing

On the continent, for all models, we find a consistent weakening of the katabatic forcing (Fig. 5b). This decrease is large on the coast in the Amundsen sea sector and in Adélie Land for MAR-CNRN, MAR-MPI and MAR-UKESM. Across all downscaled models, changes are also large and significant in the interior, even in locations where slopes are gentle.

Katabatic forcing is indeed computed as the product of the slope and the strength of the inversion layer ( $\Delta\theta$  in Eq. (3)). Here, as the surface slope does not change, the significance of changes in  $\Delta \text{KAT} = \frac{\overline{V}}{|V|} \cdot V_{\text{KAT}} (2080 - 2100) - \frac{\overline{V}}{|V|} \cdot V_{\text{KAT}} (1980 - 2000)$  (see Eq. (2.3.3)) reflects the significance of changes in the inversion strength due to Antarctic surface warming. These changes are larger in areas where the inversion strength is large at present day ( $\Delta\theta > 20^{\circ}C$ ): the high plateau and the ice shelves (Figure S5), which explains the significant changes at the center of Antarctica.

Associated with changes in  $\Delta\theta$ , the depth of the temperature deficit layer  $\hat{\theta}$  also changes. It reduces considerably on the continent, near the coastline (Figure S6), causing a reduction in thermal wind (Figure S6). Because the latter on average opposes the direction of the downslope winds (Davrinche et al., 2024), a weakening of the thermal wind increases the resulting wind speed and compensates for the decrease in katabatic acceleration. The compensating effect of thermal wind is particularly

pronounced in coastal East Antarctica where it often exceeds the decrease in katabatic forcing (Figure S7). As thermal wind and katabatic forcing both result from the forcing by the surface (SURF = KAT+THW), in the rest of the study we will call "changes in the forcing by the surface" ( $\Delta$ SURF) the changes in wind speed linked to changes in the sum of katabatic and thermal wind forcings. In general, SURF increases on the coastline and decreases elsewhere.

# 3.3.3 Changes in passive terms: turbulence, Coriolis, and advection accelerations

The contribution of horizontal advection is negligible almost everywhere, except on the Amery ice shelf. Unlike advection, the turbulent forcing is strong and encompasses surface drag. Therefore, it resembles (but with an opposite sign) changes in the sum of the dominant active accelerations. Changes in the scalar product of turbulent wind vector and the wind direction ( $\Delta$ TURB, Fig. 5f) are positive when friction decreases and negative when friction increases.  $\Delta$ TURB increases in all downscaled models over the ocean where westerlies intensify the most, decreases in the coastal margins in locations where easterlies weaken and increases overall in the interior.

To conclude, Figure 5 shows that, although surface wind changes during the  $21^{st}$  century are small on the continent, and often not consistent between models, they result from the complex interplay between changes in large-scale forcing that generally induce an increase in wind speed, and changes in the surface forcing that mostly induce a decrease in wind speed. The change in surface forcing results from a reduction in the surface temperature inversion and is consistent between models over the whole continent. However, the change in large-scale forcing varies greatly between models, with some regions of consistent changes (Adélie Land, Enderby Land, Shackleton,the Ross and Filchner ice shelves and in the Amundsen embayment region). In the following sections, we explore in more detail the regions of significant increase and decrease in wind speed across models to attribute these changes more precisely.

#### 3.4 Attribution of significant wind speed increase

For all downscaled models, in locations where the increase in wind speed by the end of the 21<sup>st</sup> century is significant, there are more than 6 times more grid cells exhibiting a significant increase in large-scale forcing than an increase in forcing by the surface pressure gradients (see Fig. 6a, 6c and Table S3). Furthermore, the proportion of significant increases in large-scale forcing is higher among grid cells exhibiting significant increases in wind speed (Fig. 6a) than in all continental grid cells. This indicates that significant increases in wind speed are likely linked to significant increases in large-scale pressure gradient forcing.

More specifically, in Adélie Land, there is a large area (denoted by a black and yellow dashed line on Fig. 7) where all GCMs agree on a significant increase in both wind speed and large-scale forcing ( $\Delta|\overline{V}| > + 0.4~{\rm m\,s^{-1}}$  and  $\Delta LSC > +0.6~{\rm m\,s^{-1}}$  for all models, see Table S4 and Fig. 7a and b). However, changes in the surface forcing are weaker (see KAT+THW on Figure 7c and  $-0.2 < \Delta SURF < 0.4~{\rm m\,s^{-1}}$  for all models in Table S4). In this specific area, changes in wind speed are well correlated with changes in large-scale forcing (R > 0.7 for all downscaled models except MAR-MPI for which R $\sim$ 0.3). The same conclusion can be drawn for Enderby Land (Figure S8 and Table S5).

**Figure 6.** Percentage of the continental grid cells exhibiting a significant (a) increase or (b) decrease in large-scale forcing or (c) increase or a (d) decrease in surface forcing in all Antarctica (black bars), among grid cell exhibiting a significant increase (red bars) or decrease (blue bars) or no change (orange bars) in July wind speed between 2080-2100 and 1980-2000. MAR-MMM indicates changes in the multi-model mean (MAR-IPSL, MAR-UKESM, MAR-MPI and MAR-CNRM) while >3M indicates significant changes observed in at least 3 GCMs downscaled by MAR.

Figure 7. Projections of 10-m changes in July wind speed in Adélie Land between 2080-2100 and 1980-2000 (a), linked to large-scale forcing (column b), katabatic forcing (column c), thermal wind forcing (column d) and total surface forcing (sum of katabatic and thermal wind, column e) for MAR-IPSL (line 1), MAR-UKESM(line 2), MAR-MPI (line 3), MAR-CNRM (line 4) and the multi-model mean of the 4 downscaled GCMs (line 5). Dotted areas indicate locations for which changes are significant at a 80 % level for the metric and the model considered. Dotted lines indicate areas for which changes in wind speed  $(\Delta|\overline{\mathbf{V}}|)$  are significant at a 80 % level for the considered model while dashed black and yellow thick lines indicate locations for which changes in wind speed  $(\Delta|\overline{\mathbf{V}}|)$  are significant at a 80 % across at least 3 downscaled models. Solid grey lines indicate elevation contours (1000, 2000 and 3000 m).

Similarly, on the Ross ice shelf (Figure S9), there is also a patch for which all GCMs project a significant strengthening of wind speed, except MAR-MPI. For MAR-IPSL and MAR-UKESM, significant increases in wind speed are also associated with significant increases in large-scale forcing (Figure S9b): on average,  $\Delta LSC > +0.9~\mathrm{m\,s^{-1}}$  for these two models, while  $\Delta SURF$  is negative (see Table S6). However, for MAR-CNRM and MAR-MPI, the increase in wind speed is not associated with any significant change in large-scale forcing ( $\Delta LSC \sim 0~\mathrm{m\,s^{-1}}$ , see Table S6) but with an increase in surface forcing ( $\Delta |\overline{V}| > +0.3~\mathrm{m\,s^{-1}}$ ) for both models, only statistically significant for MAR-CNRM (Figure S9e). Overall, on the Ross ice shelf, trends are not consistent across models for any of the forcings (Figure S9e). Although it is clear from the analysis of Adélie and Enderby Land that significant increases in the large-scale forcing drive changes in the near-surface wind speed, the analysis of Ross ice shelf (Figure S8, MAR-CNRM) indicates that surface forcing can also contribute to a significant increase in wind speed. In conclusion, significant increases in wind speed are on average more linked to significant increases in large-scale forcing but in some areas, they can also result from the changes in the surface forcing as well. Averaging over the whole continent would mask the influence of the forcing by the surface.

#### 3.5 Attribution of significant wind speed decrease

420

For all GCMs, significant decreases in wind speed are rarer (14 %) than significant increases (23 %, Table 4). Furthermore, in locations where the decrease in wind speed by the end of the 21<sup>st</sup> century is significant, there are between 1.5 (MAR-IPSL) and 14 (MAR-MPI) times more grid cells exhibiting a significant decrease in surface forcing than a decrease in large-scale pressure gradients (Fig. 6b and 6d; Table S3 and S7). This indicates that the decreases in total wind speed result from changes in the surface pressure gradients (SURF = KAT + THW) forcing. We have noted before that SURF decreases significantly in more than 30 % of grid cells, but wind speed is significantly lower in only 6 to 18 % of the grid cells. We hypothesize that the wind speed significantly decreases only when the decrease in SURF is not masked by an increase in large-scale pressure gradients, i.e., where in large-scale pressure gradients are either weak or negative.

In the Amundsen embayment region, for instance, there is an area (top left on Fig. 8, denoted by a black and yellow dashed line) where all models, except MAR-IPSL, agree on a significant decrease in both wind speed and surface forcing (Fig. 8a and e) while changes in the large-scale forcing (Fig. 8b) are weak (for MAR-MPI) to positive (MAR-UKESM and MAR-CNRM). For all continental grid cells in the Amundsen embayment region exhibiting a decrease in wind speed, changes in surface forcing are negative in all models (ΔSURF< 0.4 m s<sup>-1</sup>, see Table S8) while changes in large-scale forcings are mostly positive, except for MAR-MPI (Δ LSC = -0.25 m s<sup>-1</sup>). In conclusion, in the Amundsen embayment region, changes in surface forcing are not masked by changes in large-scale forcing and drive the decrease in near-surface wind.

Similarly, on Shackleton (Figure S10) and Filchner ice shelves (Figure S11), all models except MAR-IPSL agree on a significant decrease in both wind speed and surface forcing (Figure S11a and S11e) while changes in large-scale forcing (Figure S11c) are either positive ( $+0.5 \text{ m s}^{-1}$  for MAR-IPSL) or weaker than changes in near-surface forcings (see Table S9 and S10). Therefore, changes in surface forcing are not masked by changes in large-scale pressure gradients and drive the decrease in near-surface wind.

Figure 8. Same as Fig. 7, but for the Amundsen embayment region.

In conclusion, significant decreases in wind speed across multiple GCMs are on average more related to a significant decrease in surface forcing.

# 4 Discussion

One of the novelty of this paper was to use multiple downscaled GCMs to investigate the changes in near surface winds in Antarctica. We find that using downscaled simulations overall helps to better represent the Antarctic boundary layer processes, and thus near-surface winds, during the historical period. We are confident that this result will remain true in future projections. However, the performance of our downscaling by MAR is limited in regions of complex topography, such as the Transantarctic mountains and the interface between the coast and the southern ocean. Increasing the resolution of MAR would be computationally costly but could lead to a better representation of the surface winds in these regions. Compared to previous studies using GCMs to assess future changes in wind speed by the end of the 21<sup>st</sup> century, we were able to confirm the poleward shift and strengthening of oceanic westerlies in the MMM (Yin, 2005; Bracegirdle et al., 2008; Goyal et al., 2021). However, we show that, unlike in Bracegirdle et al. (2008) and Van Den Broeke et al. (1997), there is a significant weakening of the winter easterlies in coastal East Antarctica in the MMM. As we show that significant decreases in wind speed across multiple GCMs are on average more related to a significant decrease in surface forcing, we attribute these differences to the finer representation of the surface gradients, and thus winds, in the downscaled versions of the models.

Additionally, we performed our analysis with four downscaled GCMs instead of one, which enables a more robust analysis of of our findings. Further extending our analysis to more downscaled GCMs in the future could increase even more confidence in our results. The current number of simulations used here allows us to nuance the findings of Bintanja et al. (2014b). The latter study stated that for one GCM, climate-related (zonally averaged) wind speed changes over the continent were insignificant with respect to the interannual variability and could only be linked to changes in the large-scale forcing. We show evidence that different areas with roughly the same latitude can have opposite but significant projected changes in near-surface winds (namely, Adélie Land and Shackleton ice shelf for instance) and that these changes can originate either from changes in the surface forcing or from changes in the large-scale pattern of circulation.

As underpinned by Bintanja et al. (2014b), the results are based on simulations that can be model specific, especially with respect to the representation of large-scale circulation (Agosta et al., 2015). While changes in surface forcings are uniform across all simulations, we find that the largest inter-model differences in wind patterns originate from differences in the large-scale pressure gradients. We have also investigated the link between the strengthening of the polar cell and large-scale pressure changes at the surface in the different models but were unable to identify an obvious link between the two of them (Figure S4). The significance of changes in large-scale pressure gradients, as well as their attribution to specific mechanisms remain to be established, with an extension of this study to more models with different dynamical responses to anthropogenic warming. Last, we have performed this study with a fixed topography on the continent. Therefore, we have not assessed whether changes in both large-scale and surface forcing might be affected by changes in topography linked to dynamical loss of the Antarctic ice sheet as in Steig et al. (2015), where they show evidences of an increased cyclonic flow in regions where the topography

is reduced. Future work should be done to study the effect of changing topography on the projections of near-surface winds, large-scale and surface pressure gradients.

#### 5 Conclusions

In this paper, our first goal was to investigate the changes in near-surface winds in Antarctica, using four downscaled GCMs. Under the SSP585 scenario, in all simulations, we find a clear strengthening and poleward shift of the westerlies around Antarctica during the  $21^{st}$  century, linked to changes in large-scale forcing. GCMs with strong sea ice loss also exhibit a more pronounced poleward shift, linked to their changes in the SAM. On the continent, changes in wind speed are much weaker and with regional disparities. While all downscaled models show evidence of decreasing easterlies locally, their location vary greatly across models: in East Antarctica for MAR-UKESM and MAR-CNRM, west of Dronning Maud Land for MAR-IPSL or west of Ross ice shelf for MAR-MPI). This results in few areas of significant decrease in the multi-model mean. However, a robust feature in every downscaled GCMs is that they all exhibit a significant strengthening of near-surface wind speed in Adélie Land, on the Ross ice shelf and Enderby Land.

These patterns of change projected with MAR forced by 4 different GCMs are similar to those projected by the GCMs alone. However, when we look into the details, the GCMs alone do miss a few significant changes both on the continent and over the ocean. The decrease in coastal easterlies in all models is stronger in the MAR downscaling, where changes in the surface forcing are likely better represented. Additionally, for all GCMs, we found that downscaling with MAR significantly improves the representation of near-surface winds, except in the Transantarctic mountains and at the interface between the coast and the ocean.

Then, the second goal of this paper was to explore the drivers of these simulated changes in near-surface wind speed in Antarctica. For all GCMs downscaled by MAR, under the SSP585 scenario, the temperature inversion at the surface of the continent ( $\Delta\theta$ ) weakens (between -6% averaged over the continent for MAR-UKESM and -10% for MAR-MPI). The strongest decrease in  $\Delta\theta$  is found in the interior and on the ice shelves (Figure S5). Consequently, there is a significant decrease in the katabatic forcing, consistent across all downscaled GCMs, in coastal regions and in the interior as well. Simultaneously, due to warming of the surface, the ability of coastal margins to accumulate cold air at the foot of the slope is reduced (Figure S6d). Therefore, we also observe a significant weakening of thermal wind forcing in coastal areas. Because the thermal wind opposes the dominant direction of the downslope winds in the sloped regions of Antarctica  $\sim$  250 km from the coastline (Davrinche et al., 2024), a weakening of the thermal wind forcing increases the resulting wind speed and compensates for the decrease in the katabatic acceleration in these onshore regions. The compensating effect of thermal wind is particularly pronounced in coastal East Antarctica where it often exceeds the decrease in katabatic forcing, leading to an overall increase in the wind speed resulting from the surface forcing only. The changes in large-scale forcing are spatially less uniform and less consistent across models, but it overall exhibits larger areas of significant increases than decreases.

From our statistical analysis and case studies, we conclude that (i) significant decreases in wind speed are statistically more linked to changes in surface forcing, when not masked by an increase in large-scale forcing (as shown on Shackleton, in the

Amundsen embayment region and on Ross ice shelves), and (ii) significant increases in wind speed are statistically more linked to changes in large-scale forcing (as shown in Adelie, Enderby Land and Filchner ice shelf).

This work paves the way for more studies exploring the impacts of the above-mentioned changes in near-surface winds. For example, changes in the mean value of winter near-surface wind speed are likely to impact the quantity of drifting snow and sublimation, and the stability of ice shelves through potential enhanced surface melt (Lenaerts et al., 2017). We expect sublimation and drifting snow to be reduced in case of a weakening of the wind speed. However, further studies should be performed to quantify these effects.

Code and data availability. All Codes and dataset to analyze future changes in near-surface winds in Antarctica under the SSP585 scenario are available at https://zenodo.org/records/14191007. Data of the AntAWS are available from Wang et al. (2023) (https://doi.org/10.48567/key7-ch19).

Author contributions. CD, CeA, and AO designed the study and contributed to the output and observation analyses. CeA, ChA, and CK set up the MAR model for Antarctica with several adaptations. ChA, CK, CeA and CD performed model simulations. CD performed the momentum budget decomposition, post-processed the data, did the bulk of the analysis, and made all the figures. CD wrote the first draft, with input from CeA and AO. All authors contributed to discussions in writing this paper.

Competing interests. The contact author has declared that none of the authors has any competing interests.

Acknowledgements. This publication was funded by the ANR-JCJC Katabatic project (ANR19-CE01-0020-01) to AJO and NSERC Discovery grant DGERC-2021-00213. The authors appreciate the support of the University of Wisconsin-Madison Automatic Weather Station Program for the data set, data display, and information, NSF grant number 1924730. We acknowledge using data from the CALVA project and CENECLAM and GLACIOCLIM observatories. This work is part of the AWACA project that has received funding from the European Research Council (ERC) under the European Union's Horizon 2020 research and innovation programme (Grant agreement No. 951596), and part of the POLARiso project that has received funding from the European Union's Horizon 2020 research and innovation programme under the Marie Skłodowska-Curie grant agreement No 841073. The MAR simulations were performed thanks to granted access to the HPC resources of IDRIS under the allocations 2022-AD010114000 made by GENCI. We acknowledge the work of Xavier Fettweis (Université de Liège) in developping and mantaining the MAR model

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
