# Peer review of "Future changes in Antarctic near-surface winds: regional variability and key drivers under a high-emission scenario"

_EGUsphere, 2025_

## Author Comment (AC1)

**RESPONSE TO ANONYMOUS REVIEWER #2**

REVIEW OF DAVRINCHE ET AL., 2025 – FUTURE CHANGES IN ANTARCTIC
NEAR-SURFACE WINDS: REGIONAL VARIABILITY AND KEY DRIVERS UNDER A
HIGH-EMISSION SCENARIO

We thank the reviewer for their valuable and helpful comments on the manuscript. We propose to implement the following changes in a revised version.
Black = reviewer comment / Blue = author's response / *Italic* = revised text.

The manuscript focuses on projections of surface wind speeds over Anarctica, which is comparatively under researched. The approach is very comprehesive (e.g., four downscaled GCMs, as well as a budget analysis) and the figures are well made - and clearly a lot of effort has gone into this work. Additionally, in general much of the results are well explained, with a good level/balance of detail. So there is much merit to the paper.

However, unfortunately the manuscript is let down by other instances of poor writing and organisation - and indeed I would even go as far as suggesting that the manuscript in its present state was not ready for submission. These concerns are especially evident in the Introduction and Methods section, which come across as rather muddled, disorgansied, and disjointed. I am sure this is not reflective of the authors abilities and knowledge, so this really has to be remedied before the manuscript can be considered for publication. I would really suggest a very thorough rewrite/revision of many of the section is necessary, with all authors contributing.

Major comments:

1. Many parts of the manuscript come across as rather unpolished and the writing disjointed. This really needs to be improved.

We are taking this comment into account, and will make sure to polish the next version of the manuscript.

2. For example, many of the sentences in the Introduction claim something but do not include a citation for evidence. So sentences such as 'On the one hand, the greenhouse warming causes an increase of the incoming longwave radiation.'

We will replace this sentence as follows: "*On the one hand, the increase in GHG concentrations causes an decrease of the outgoing longwave radiation (Mitchell, 1989). As a consequence, the temperature inversion and thus the katabatic forcing should decrease (Van den Broeke and van Lipzig, 2002; Bintanja et al., 2014b).*"
In general, we will rewrite the introduction with careful attention to the inclusion of citations for any idea described.

3. and 'Although there is a consensus on the reduction of surface forcing in climate projections'.

*We will add the following references in the revised version: "(Van den Broeke and van Lipzig, 2002; Bintanja et al., 2014b)."*

4. There are also incidences of repetition, such as in the Introduction with something along the lines of 'which is proportionate to the strength of the temperature inversion' mentioned twice

*We have spotted a repetition line 34 and will remove it in the revised version.*

5. and in the methods and Introduction which both mention something along the lines of 'Because of their resolution, GCMs are not expected to perform well in locations with complex topography.'

*We did not mention specifically the resolution of GCMs in the introduction, but rather their ability to take into account complex topography, land–sea contrasts, boundary-layer and convective processe (L51).*

6. Other instances are the preambles/motivation before the results, which just say in a slightly different fashion what was said before. Please remove all repetition, and remember that your audience/readers only need to be told something once.

*We were following the article writing guidelines developed in Plaxco, 2010: "The first sentence of each paragraph should tell the reader what you expect them to get out of the paragraph that follows, which makes their job of following it far easier. Put another way; use the opening sentence of your paragraph to state your argument, and the rest of the paragraph to make your argument.". However, we will take into account your comment and will make our best to remove all opening sentences.*

7. Also there are typos, such as '(e.g. north of Ross and Amery ice shelves and north of the Peninsula' in the Introduction (so no closure of parentheses).

*We apologize for this mistake, it will be corrected in next version.*

8. Mistakes such as AWS defined, and the phrase automatic weather station still used.

*We have spotted two AWS definitions: l66 in the preamble and l 71. Because we will remove the preamble in the revised version, there will no longer be a repetition.*

9. Very random / ad hoc approaches such as using m/s in one sentence and km/hr in the following sentence (methods). These give the feel of a rushed writing process, and of a manuscript submitted before it was really ready.

We have spotted this line 84, and modified it in the revised version of the manuscript.

10. There are also parts of it which are disorgansied, such in section 2.1 mentioning ERA5, and then ERA5 not being explained until later (also it's not explained in a logical fashion from the methods that ERA5 is being used to select the GCMs.).

In Sec 2.1, ERA5 was mentioned but not explained in the preamble. Because we are removing all preambles in the revised version, ERA5 won't be mentioned before being explained.

11. Poor paragraph structure such as section 2.1.2.

We have updated this paragraph: "*ERA5 is the latest reanalysis produced by the European Centre for Medium-Range Weather Forecasts (Hersbach et al., 2020). Its horizontal spatial resolution is $\sim$ 31 km and outputs are given at a hourly frequency. The assimilation system (IFS Cycle 41r2 4D-Var) uses 10 members to produce a 4D-Var ensemble of data assimilation (Hennermann and Guillory, 2019). Among various reanalysis products (MERRA-2, JRA-55, ERAI, NCEP2, and CFSR), ERA5 has been shown to perform best in capturing monthly averaged wind speeds (Dong et al., 2020).* "

12. Finally, some odd sentences such as 'We focus on the Antarctic continent, which is the source region of the katabatic forcing' in the final paragraph of the Introduction.

This will be changed in the revised version:
L58: "*We focus on the Antarctic continent,* **where katabatic winds are developing in the sloped regions due to the quasi permanent radiative cooling by the ice sheet (Phillpot and Zillman, 1970)***, and on the winter season, as it is the season for which both the katabatic forcing and the mean wind speed are the highest (Davrinche et al., 2024).*"

13. Methods: Out of the blue it is mentioned that the subset of AWSs are selected based on their ability to represent ERA5. This is not justified. Additionally, this seems a rather strange choice, as ERA5 would also struggle to represent steep coastal gradients, so also do poorly representing katabatic winds. So justification is clearly required.

The subset of AWS is not selected based on their ability to represent ERA5. On the contrary, we show that ERA5 is not able to reproduce correctly surface wind

speed in some locations with complex topography. As we do not expect GCMs to perform better than the reanalysis over the period of available AWS observations (as stated L119), we decided to exclude AWS that were already misrepresented in ERA5, which assimilates observations in Antarctica. In the end, we exclude stations located in the Transantarctic mountains and at the interface between the continent and the ocean, which follows expectations. However, we wanted to use a rigorous method to exclude those stations.

We understand that some sentences in the manuscript might suggest that we exclude stations based on their ability to represent ERA5 instead of based on the ability of ERA5 to represent the climatology of the AWS. The following changes will thus be implemented in the revised version:

- L68: *Therefore, we **exclude** a subset of AWS based on i) the ERA5 reanalysis' **in**ability to represent the mean wind speed and variability in areas of complex topography, and ii) the length of available winter time series to evaluate GCMs on a representative climatic time scale.*
- L113: The title *"2.1.4 Exclusion of sites near complex topography based on performance of ERA5"* will be changed to *"2.1.4 Exclusion of sites near complex topography"*
- L113: *"These four stations exhibit the largest biases in terms of temporal variability ($R < 0.3$ and $\sigma N > 2$, which indicates that the variability in ERA5 is underestimated) and mean amplitude ($B > 30\%$, which indicates that ERA5 overestimates the mean value of the wind speed). **Additionally, these stations are all located at the foot of the Transantarctic mountains (Fig. 1), which justifies their exclusion in the quantitative analysis.**"*

13. The correction to the AWS dataset is also poorly explained (Equations 1 and 2) – its not even clear what is being corrected, and what 1-3 and 1-6 refers to.

We will explain more clearly what equations 1 and 2 refer to:

L76: *"According to the logarithmic theoretical profile of wind speed in the boundary layer, with a constant roughness length $z_0 = 1$ mm (Vignon et al., 2017), we estimate the maximum correction between wind speed measured at the real height of the sensor and wind speed at 3m to be between -10 % (**for the correction from 1 to 3m**) and 7 % (**for the correction from 6 to 3m**) of the theoretical value:*

$$correction_{6m-3m} = \frac{log(\frac{6}{z_0})}{log(\frac{3}{z_0})} = 1.07 \tag{0.1}$$

$$correction_{1m-3m} = \frac{log(\frac{1}{z_0})}{log(\frac{3}{z_0})} = 0.90 \tag{0.2}$$

*"*

14. Selection criteria for GCMs: This seems to state that their performance in the Arctic is also taken into account, which is completely unjustified.

For practical reasons, we did not want to use the entire CMIP6 range of models in our study. We aimed to use a small subset of 4-5 models that 1) are not too wrong, and 2) represent a range of possible climate projections. Among CMIP6 models, we have selected those that performed the best in the Antarctic. As both poles share common physical processes, we have selected among models that performed well in the Antarctic those who also performed well in the Arctic to get to a reasonable number of models. Additionally, the choice of these four models for our study is supported by another study by Williams et al. (2024) where these models were classified among the best performing ones in Antarctica, in winter when comparing their sea ice extent, surface air temperature, zonal wind at 850 and 50 hPa to ERA5.

15. There are a lot of locations mentioned, but I don't think they are always shown on a map.

All stations are shown on Fig. 1. Regions of interest are shown on Fig.4. However, the Transantarctic mountains and the Plateau are not explicitly shown on a map. We will add them on Fig. 1 in the revised version of the manuscript.

[Figure]

FIGURE 1. Elevation, from Bedmachine (a) over all Antarctica, (b) zoomed on the black rectangle area. Superimposed are the 28 pre-selected AWS. Stations that have been discarded because of the inability of ERA5 to properly represent winds at these locations (see Sect. 2.1.4) are underlined. Red dots indicate the Transantarctic mountains.

16. Minor comments (this is just a selection as there are a lot of 'minor' concerns that need to be addressed by a very thorough revision of the paper)

We will pay extra attention to re-reading and correcting all typos in the man-uscript.

17. Abstract: Not clear what the distinction between katabatic and thermally driven winds is. I think some explanation of the term 'thermally driven' is nec-essary here, as otherwise the reader is lost.

We have added some explanations in the abstract:

L11: *"These drivers include local forcings related to the net radiative cooling by the iced surface as well as large-scale forcing.* **We distinguish two types of local forcing: katabatic forcing (linked to the presence of a slope) and thermal wind forcing, which arises from horizontal gradients in the depth of the radiatively cooled surface layer.** *"*
Additionally, we have changed the sentence in the introduction to: L30: *"On the other hand, the surface forcing includes a gravitational katabatic pressure gra-dient that is proportional to the strength of the temperature inversion and a local thermal wind created by horizontal gradients in the depth of the temperature deficit layer that acts to replenish the pressure low created by the downslope displacement of air."*

18. The introduction mentions one mode of variability, the SAM. But what about the Amundsen Sea Low?

We have mentioned SAM as it is the dominant mode of variability in the southern hemisphere (Thompson and Salomon, 2002). However, we acknowledge that we should also have mentioned the influence of ENSO. We will add in the revised version of the paper:
L26: *"Large-scale forcing is intrinsically linked to the leading modes of variabil-ity in the Southern Hemisphere: the Southern Annular Mode (SAM) and the El Niño–Southern Oscillation (ENSO). The SAM is quantified by the SAM index, which represents the zonally averaged sea-level pressure gradient between 40°S and 65°S (Marshall et al., 2003). ENSO is characterized by the Southern Oscillation Index (SOI), based on sea-level pressure differences between Tahiti and Darwin (Bromwich et al., 2004). Both SAM and ENSO influence the strength and posi-tion of the Amundsen Sea Low (a persistent low-pressure center in the Amundsen Sea sector(Raphael et al., 2016)), which in turn modulates the frequency and tra-jectories of cyclones in the area (Fogt et al., 2012)."*

L35: *"On the other hand, the increase in GHG concentration drives the SAM towards a more positive phase by the end of the 21st century (Miller et al., 2006; Fogt and Marshall, 2020; Goyal et al., 2021) while the effect on the SOI remains highly uncertain (Beobide-Arsuaga et al., 2021; Ren and Liu, 2025). "*

19. The katabatic winds are also dependent on the size of the slope.

Yes, we did not mention it in this sentence, because we wanted to highlight the dependence of the katabatic acceleration to the temperature inversion, as the slope does not change between the two time periods, but we agree that it might be misleading for the reader, and will add it in the revised version: L39: " *On the other hand, the surface forcing includes a gravitational katabatic pressure gradient that is proportional to the strength of the temperature inversion and the slope angle.*"

20. Section 2.1.3: Not sure why the comment on the length of observations in the summer season is necessary.

Yes, indeed, it was a mistake, we were referring to **austral winters**. This will be corrected in the revised version.

21. And shouldn't the number of AntAWS stations mentioned here, actually be mentioned in section 2.1.1. Comes across as disorganised.

The minimum number of observations is justified based on the variability of near-surface winds derived from ERA5. Consequently, this discussion cannot be included in Section 2.1.1, as it precedes the introduction and description of ERA5. However, we have not mentioned the total number of AWS in Section 2.1.1 and will add it: L 73: " *For all **267** stations (except Zhongshan) [...]*"

22. Section 2.1.4: At least the third time that GCMs issues over representing complex orography has been mentioned. Repetition. Makes the manuscript look extremely disorganised and amateurish.

We have removed a repetition by discarding the preamble. The inability of GCMs to represent complex topography is now only mentioned once.

23. Line 118. Typo. Grill -¿ Grid

Yes, this will be corrected in the revised version.

24. Methods: Its not clear what the term 'Implausibility' is being used here for.

We understand this comment and will give more details in the revised version: " *Fraction of implausibility*" *is defined for each metric as the portion of the surface where the difference between historical averages in the model and ERA5 is greater than a plausible threshold set at 3 times the ERA5 interannual standard deviation (Agosta et al., 2022).*"

25. Section 2.3.2: Poor paragraph structure.

26. Section 3.1: The preamble here is inappropriate / repetition. This material should be in the Introduction or Methods, not repeated at the beginning of the

results section. This weakens the paper and makes it look disorganised.

The preamble here will be removed in the revised version. Parts of the preamble have been moved to L49 in the introduction.

27. Section 3.3: Similar comment to above, no need for the preamble.prevalence

The preamble will be removed in the revised version.

28. Line 319: SAM defined again

Yes, it is indeed the second time the SAM is defined. We will remove the bracket.

29. Section 3.3.1: Huge amount of repetition on how SAM will change.

We will thoroughly rewrite the paper to avoid unnecessary repetitions. However, it is also a matter of style and efficiency to have a few repetitions of key concept in a 25 pages manuscript. The SAM trends are described in 1 sentence in the introduction

> "*The increase in GHG concentration drives the SAM towards a more positive phase by the end of the $21^{st}$ century (Miller et al., 2006; Fogt and Marshall, 2020; Goyalet al., 2021)*"

and then once in the results:

> "  *This result is in agreement with previous studies that showed that the already observed increasing positive trend of the SAM will likely continue in response to increasing greenhouse gases and after the recovery of the ozone hole (which offsets the strengthening of the SAM (Bracegirdle et al., 2008))*."

We checked for repetitions in the paper and have removed most of them. But we we would like to keep the sentences on how SAM will change as we consider that 2 mentions, 16 pages apart, does not harm the flow of the paper.

**1. REFERENCES**

Beobide-Arsuaga, G., Bayr, T., Reintges, A., & Latif, M. (2021). Uncertainty of ENSO-amplitude projections in CMIP5 and CMIP6 models. Climate Dynamics, 56, 3875-3888.

Bromwich, D. H., Monaghan, A. J., & Guo, Z. (2004). Modeling the ENSO modulation of Antarctic climate in the late 1990s with the Polar MM5. Journal of climate, 17(1), 109-132.

Fogt, R. L., Wovrosh, A. J., Langen, R. A., & Simmonds, I. (2012). The characteristic variability and connection to the underlying synoptic activity of the Amundsen-Bellingshausen Seas Low. Journal of Geophysical Research: Atmospheres, 117(D7).

Mitchell, J. F.: The "greenhouse" effect and climate change, Reviews of Geophysics, 27, 115–139,600 https://doi.org/10.1029/RG027i001p00115, 1989.

Phillpot, H. R., & Zillman, J. W. (1970). The surface temperature inversion over the Antarctic continent. Journal of Geophysical Research, 75(21), 4161-4169.

Ren, X., & Liu, W. (2025). Distinct anthropogenic aerosol and greenhouse gas effects on El Niño/Southern Oscillation variability. Communications Earth & Environment, 6(1), 24.

Plaxco, K. W. (2010). The art of writing science. Protein science: a publication of the Protein Society, 19(12), 2261.

Raphael, M. N., Marshall, G. J., Turner, J., Fogt, R. L., Schneider, D., Dixon, D. A., ... & Hobbs, W. R. (2016). The Amundsen Sea low: Variability, change, and impact on Antarctic climate. Bulletin of the American Meteorological Society, 97(1), 111-121.

Roussel, M. L., Lemonnier, F., Genthon, C., Krinner, G. (2020). Brief communication: Evaluating Antarctic precipitation in ERA5 and CMIP6 against CloudSat observations. The Cryosphere, 14(8), 2715-2727.

Thompson, D. W., & Solomon, S. (2002). Interpretation of recent Southern Hemisphere climate change. Science, 296(5569), 895-899.

---

## Author Comment (AC2)

**RESPONSE TO ANONYMOUS REVIEWER #1**

**REVIEW OF DAVRINCHE ET AL., 2025 – FUTURE CHANGES IN ANTARCTIC NEAR-SURFACE WINDS: REGIONAL VARIABILITY AND KEY DRIVERS UNDER A HIGH-EMISSION SCENARIO**

We thank the reviewer for their valuable and helpful comments on the manuscript. We propose to implement the following changes in a revised version.

Black = reviewer comment / Blue = author's response / *Italic* = revised text.

Antarctic near-surface winds are important for blowing snow, precipitation and ice shelf stability. This article presents projections of Antarctic winds for the end of the century under a high-emissions scenario, using the MAR model with a suite of driving global models from CMIP6. A comprehensive overview of changes in winds is presented, with meticulous evaluation of the added value of dynamical downscaling. The authors also decompose the momentum budget in the regional model projections. Although a previous paper on future winds (Bintanja et al., 2014) did estimate two of these terms, to my knowledge this is the first to provide a full budget decomposition of projected winds from regional model output. The results are novel and interesting and I would support this paper's publication subject to revisions.

**Major comments**

The main points I raise in this review are regarding the application of the budget:

1. It's not quite clear from this analysis how the momentum budget decomposition was applied for each of the model configurations. Was your choice of parameters for diagnosing theta0 the same as in Davrinche et al. (2024)? Is there a way to test how robust your results are to the choice of Hmin?

For each model downscaled by MAR, we have performed the momentum budget decomposition following Davrinche et al. (2024). In this previous paper, in the supplement, we have validated our method and tested how robust our decomposition was to the choice of Hmin. We have also evaluated our pressure gradient force (PGF = KAT + LSC + THW) from the momentum budget decomposition (PGF) against the native pressure gradient force, which is a native output from MAR. It is also equivalent to comparing the native turbulence with the residual term. Because a lot of effort has been dedicated to the evaluation of the method in Davrinche et al., 2024, we are not planning on adding too much information about the evaluation in this paper. However, as mentioned in Point 15, we have added a quantification of the error arising from closing the momentum budget. We will also update the following sentence in the revised version:

L190: "*The method is described extensively in Davrinche et al. (2024).* **For each model downscaled by MAR***, we compute the momentum budget in the cross- and downslope directions and we decompose it into 6 different accelerations,defined as follows:*"

2. The regional analysis is quite long and has a large figure count – you may be able to significantly improve readability by reducing the number of figure panels shown in Section 3.5, which focuses on a very specific question (the drivers of decreases). The vast majority of the changes shown in those regional panels are already visible in Figure 5.

We thank the reviewer for this suggestion. We will take it into account by moving figure 8 and 10 to the supplement.

3. A time-correlation analysis between the wind speed and budget terms would really help understand the role that surface forcing plays, instead of just looking at the change. How much variance in the July monthly mean between 1980 and 2100 do we explain with just VLSC from Equation 5, then how much when we add other terms?

We refer the reviewer to Section 4.4 of Davrinche et al., 2024 for a detailed study of the correlation between wind speed and the momentum budget terms. Given how long this paper already is, we did not want to repeat that analysis here. In this previous paper, we looked at the drivers of near-surface variability at the seasonal and 3-hourly timescale. Figure 1 and 2 (Figure 8 and 10 in Davrinche et al., 2024) are of specific interest to illustrate the role that surface plays in explaining the variance of July wind speed. We showed that the correlation between large-scale acceleration and total wind speed is high in locations where katabatic acceleration is weak. However, closer to the coast, none of the katabatic nor large-scale accelerations alone controls the variability at the 3-hourly timescale.

[Figure]

FIGURE 1. Seasonal cycle of 3-hourly nea-surface winds averaged over 10 years for (a) total wind speed, (b) wind speed equivalent to large-scale acceleration, (c) wind speed equivalent to thermal wind, (d) wind speed equivalent to advection, (e) wind speed equivalent to horizontal katabatic and (f) wind speed equivalent to turbulent accelerations. Note that the y-axis is different between panels a-d ($|WS|$, $|V_{LSC}|$, $|V_{THWD_{TD}}|$, $|V_{ADVH}|$) and panels e-f ($|V_{KAT}|$, $|V_{TURB}|$).

[Figure]

FIGURE 2. (a) Average July 2010-2020 correlation coefficient of 3-hourly katabatic acceleration and wind speed (b) Average July 2010-2020 correlation coefficient of 3-hourly large-scale acceleration and wind speed (c) directional constancy of 3-hourly large-scale wind speed. (d, e, f): Mean of 3-hourly July 2010-2020 scalar product normalised by the norm of wind speed of (d) 3-hourly katabatic wind speed and total wind speed, (e) 3-hourly large-scale and total wind speed, (f) 3-hourly thermal-wind and total wind speed, (g) 3-hourly advection and total wind speed. For the 7 panels, the dotted black line corresponds to the line for which the correlation coefficient of katabatic acceleration and total wind speed reaches 0.5. Seven zones of higher correlations are indicated: (I), (II), (III), (IV), (V), (VI) and (VII)

On longer timescales, on the continent, the large scale forcing dominates inter-annual variability (Fig. 3), while changes in near-surface forcing result in lower frequency variability. Figure 3 also shows that the dominant drivers identified on the 1980-2000 period are still the same over 2080-2100. Here, we wanted to add to the previous paper by looking specifically at the change in the different terms between the two time periods, rather than to directly look at the 20 years time correlations.

We are also limited by the fact that the momentum budget decomposition is computationally heavy, and we could not run it for 100 years, but for 2 times 20 years. We are in essence comparing two time slices, and cannot do a statistical analysis on just two numbers, showing the difference, as we did, is more appropriate.

As we understand that we cannot expect all readers to have thoroughly read Davrinche et al., 2024, we will add some key results in the revised version of the manuscript:

l32: "**At present day, large-scale forcing dominates the variability of wind speed on the interior, while closer to the coast, none of the katabatic nor large-scale accelerations alone controls the variability**

*at the 3-hourly timescale (Davrinche et al., 2024). In future projections, [...]"*

[Figure]

FIGURE 3. Proportion of variance of continental monthly July wind speed explained by the different accelerations for July 1980-2000 (black) and July 2080-2100 (green).

**Specific comments**

4. L30: A "thermal wind that acts to replenish the pressure low created by the downslope displacement of air". Could you describe what you mean by this in more detail – a thermal wind operates at large scales in a baroclinic atmosphere.

Do you mean the local thermal wind acceleration term or the thermal wind relationship used to calculate lsc?

*We meant the thermal wind induced by horizontal gradients in the depth of the temperature deficit layer. It is explained in more details in Sec. 2.3.2. It quantifies the effect of baroclinicity at low levels. As we understand that it might confuse the reader in the introduction, we will rephrase:*
*L30: "On the other hand, the surface forcing includes a gravitational katabatic pressure gradient that is proportional to the strength of the temperature inversion and a local thermal wind created by horizontal gradients in the depth of the temperature deficit layer that acts to replenish the pressure low created by the downslope displacement of air."*

5. Intro: please briefly review Bintanja et al. (2014) and consider the advancements made here relative to that research. I see you reference them later – it's worth signposting early on.

*The reference will be mentioned earlier in the introduction: L60 "We mitigate GCM limitations used in previous studies (Bintanja et al., 2014) by downscaling them with using the regional atmospheric climate model MAR. We use the momentum budget decomposition to analyse how each family of drivers evolves in the different downscaled GCMs. In addition, we perform this analysis for four recent CMIP6 GCMs carefully selected on their ability to represent the large-scale circulation in polar regions. It enables us to mitigate single-model analysis issues and to test how robust potential changes are."*

6. L54: I'm not sure if dynamical downscaling alone ensures a physically realistic simulation of boundary-layer dynamics. Rephrase perhaps?

*Yes, it is not just the downscaling, but also the better model physics over snow in MAR that leads to improvements. We will rephrase it in the revised version: "This ensures a better resolution of the ice sheet topography as well as a more realistic simulation of boundary-layer dynamics achieved through adapted parametrizations of the interactions between the snow/ice surface and the atmosphere."*

7. L100: Why select July and not the more usual climatological season of JJA?

*The cost of computation and storage of the momentum budget terms for the four downscaled models is high and we could not afford saving all variables. Hence the limited number of studied months.*

8. The supplement is very large and there's a lot of flipping back and forth between the main text and the supplement. If there is a way you can reduce the size of the supplement it would improve the flow. S1 and S2 are one equation I believe?

We feel that removing parts of the supplement would weaken our analysis. However, in order to improve the flow, we will remove as many back and forth flipping as possible. For instance:

- L399: As Ross ice shelf is now presented in the supplement, this has removed a supplement-main body flip
- L426: As Shackleton ice shelf is now presented in the supplement, this has removed a supplement-main body flip

9. S1.1: is the relative uncertainty the standard error?

Not exactly, the relative uncertainty is the standard error divided by the mean value:

$$\text{Relative uncertainty} = \frac{\frac{\sigma_{July}}{\sqrt{N_{July}}}}{|\overrightarrow{V_{July}}|} \tag{0.1}$$

We acknowledge that it might be complicated for the reader, and will update this sentence in the revised version of the manuscript:
Supplement, l3: *"To test whether datasets are long enough to be representative of a climatological period, we compute using ERA5 the minimum value of $N_{July}$ for which the standard error on the mean value of the July wind speed between 1980 and 2020 is inferior to 5 % of the mean value."*

10. L124: why calculate the metrics for December and the annual mean if we are only focused on July?

We wanted to give a more general result regarding the added value of downscaling by MAR regarding the representation of near-surface winds. We understand that it is not the primary focus of this paper, and will therefore present only July in the main body:

- We will compute the TPS for July only, and will move the last 6 colums of Table 4 to the supplement.
- We will update Fig. S3 as follows:

[Figure]

FIGURE 4. Score of the 28 pre-selected AWS stations compared to ERA5 for all July available AWS data Three metrics are considered: the correlation coefficient (R), the normalized mean bias (B) and the normalized standard deviation ($\sigma_N$). Each metrics for each station gives a score equal to -1 and 1 depending on its performance (see Sec. 2.1.4). Positive values indicate a good performance. (a) Scores for each metrics and for each stations. (b) Sum of all individual scores. Red solid line on the colorbar indicates the threshold under which stations are excluded based on their comparison with ERA5. Those stations are shaded in blue.

- Update underlined stations on Fig. 1
- Update Table 2
- Update Fig. 3 as follows:

- We will update Sec. 3.1
- We will remove Fig. S4

11. Supplement L16: check the reference to 'Figure 5 in the manuscript'.

It should indeed refer to Figure 3, thit will be updated in the revised version of the manuscript.

12. Supplement: in my PDF Figure S3 shows after Figure S4.

It will be corrected in the revised version of the manuscript.

13. Figure S3: I am a bit confused by the (b) panel colour bars. What is the left and right coloured bar showing? Maybe it would be simpler to show each

[Figure]

FIGURE 5. (a) Altitude of the selected stations. Mean normalized bias (B) for wind speed with regard to the AntAWS observations (B $= (|\overrightarrow{\overline{V_{GCM}}}| - |\overrightarrow{\overline{V_{AntAWS}}}|)/|\overrightarrow{\overline{V_{AntAWS}}}|$ for (a) and B $= (|\overrightarrow{\overline{V_{MAR-GCM}}}| - |\overrightarrow{\overline{V_{AntAWS}}}|)/|\overrightarrow{\overline{V_{AntAWS}}}|$ for (b)) for the 24 selected AntAWS stations, computed for July (b) using the GCMs, (c) using the GCMs downscaled by MAR.

individual TPS on the grid and e.g. hatch the gridcells which pass the threshold?

As mentioned above, we will update Figure S3 and replace the heatmap by a regular map with blue shaded stations indicating stations that do not pass the threshold.

14. L203 Is this strictly speaking the boundary layer? You imply here that the height at which the vertically integrated temperature deficit becomes zero is the top of the boundary layer. In East Antarctica however the temperature deficit can extend to 4km height (see e.g. Figure 3 in van den Broeke and van Lipzig, 2003). This is much deeper than the top of the stable boundary layer, which vdB and vL say is 'poorly constrained'. Is it not more correct to say that it's just the vertical integral of the temperature deficit? My understanding is that the temperature deficit can extend far above the boundary layer, which over the plateau may be e.g. 10-150m at Dome C, Pietroni et al., 2012: https://doi.org/10.1007/s10546-011-9675-4

Yes, it is correct. It will be mentioned in the revised version of the manuscript: "$\hat{\theta}$ is the vertically integrated potential temperature deficit from the top of the inversion layer. Above the inversion layer, as $\theta = \theta_0$, both $\Delta\theta$ and $\hat{\theta}$ become zero."

15. L226 does the residual here also encompass any errors from closing the budget (e.g. finite difference approximations) or is it directly output from the model?

Yes, we will detail that L226: *"The residual term (**TURB**) encompasses vertical advection (which is weak), turbulent drag (which opposes the other accelerations and is strong when the wind speed is high)* **and potential errors arising from closing the momentum budget. A comparison of MAR's native turbulent acceleration and our recomputed residual turbulence as detailed in Davrinche et al., 2024 enables us to conclude that the error resulting from closing the budget in July is weak compared to the absolute value of the turbulence (ie $\sim$ 10 % for all models).**"

16. Figure 3a – what is the x-axis here

The former x-axis represented the stations' number. We removed these values to only keep the ticks (see Fig. 5).

17. Section 3.1 no need to restate this first para, or move to the introduction.

This paragraph will be moved to the introduction in the revised version.

18. L270 figure panel reference needed

The panel reference will be added in the revised version.

19. Figure 3: I think I missed what the collocation method is? Are you using nearest neighbour or bilinear? Is MAR regridded to the same grid as the ESMs? If not it would be useful to do this as an additional analysis to just check if the added value comes from being able to collocate a gridpoint closer to the location of the AWS in MAR.

We forgot to mention it, but we did regrid the GCMs on MAR's grid using a bilinear interpolation. We will add it in the revised version:
*"They are regridded using a bilinear interpolation on MAR's grid."*

20. Figure 4: (v) not quite able to tell but it looks like this is not the Ronne ice shelf? It may be worth checking – in my understanding the Ronne hugs the peninsula and the Filchner ice shelf is east of that.
Yes, according to this detailed map (), region (V) is closer to Filchner ice shelf than to Ronne ice shelf. This will be modified in the figures and in the main body of the revised manuscript.
21. Table 3: in my PDF this appears below Figure 4 (but referred to beforehand).

It will be corrected in the revised version.

22. L357: my understanding is this ^ is the vertical integral of the deficit rather than the depth of the layer

Yes, it would be more correct to state that :
"*Associated with the changes in $\Delta\theta$ at the surface, the depth of the temperature deficit layer also decreases. Therefore, $\hat{\theta}$ reduces considerably* **on the continent**, *near the coastline (Figure S7), causing a reduction in thermal wind (Figure 5d).*"

23. L357: please specify where in Figure S7 you are referring to for the coastline

As stated in the Point 22, we will add the group of words "**on the continent**", as we are focusing on onshore winds.

24. L359: in some regions (e.g. offshore of Adelie land) the thermal wind is a positive forcing term and does not oppose the katabatic wind so it doesn't necessarily increase wind speed if you reduce it.

We were focusing on onshore winds but forgot to mention it in this sentence. As stated in the points 22 and 23, we will add the group of words "**on the continent**".

25. L398: where are these regions where 'surface forcing can also contribute to significant wind speed increase'?

In L398, it is specified for "Ross ice shelf", but we should have mentioned the MAR-CNRM model. In this simulation, the large-scale acceleration decreases, and the observed significant increase in total wind speed can only be linked to changes in the surface forcing. We will change the following sentence accordingly: "*L398: While it is clear from the analysis of Adélie and Enderby Land that significant increases in the large-scale forcing drives changes in the near surface wind speed, analysis of Ross ice shelf (Fig. 8, MAR-CNRM) indicates that surface forcing can also contribute to significant wind speed increase.*"

26. L455 you imply here that some regions have an increased wind speed due to surface forcing, and it's true that the surface forcing does increase (kat+thw) in some regions but I don't see these mapping onto obvious increases in wind speed.

What we meant here was that wind speed resulting ONLY from surface forcing was overall increasing. "*L455: Because the thermal wind opposes the dominant direction of the downslope winds (Davrinche et al., 2024), a weakening of the thermal wind forcing increases the resulting wind speed and compensates for the decrease in katabatic acceleration. The compensating effect of thermal wind is particularly pronounced in coastal East Antarctica where it often surpasses the*

*decrease in katabatic forcing, leading to an overall increase of the wind speed **resulting from** the surface forcing.*"

28. PIG -> Amundsen embayment region?

This will be updated in the revised version.

29. L454 regional specifics would be helpful here as this compensating effect only applies where the katabatic winds are active

Yes, it is true. As we do not want to introduce new categories of regions depending on the elevation or slope (as we did in (Davrinche et al., 2024)), we will add some descriptions in the sentences instead: "*L454: Because the thermal wind opposes the dominant direction of the downslope winds **in the sloped regions of Antarctica $\sim$ 250 km from the coastline** (Davrinche et al., 2024), a weakening of the thermal wind forcing increases the resulting wind speed and compensates for the decrease in katabatic acceleration **in these onshore coastal regions**. The compensating effect of thermal wind is particularly pronounced in coastal East Antarctica where it often surpasses the decrease in katabatic forcing, leading to an overall increase of the wind speed due the surface forcing **only**.*"

30. Section 4: I may have missed it but I think the added value of dynamical downscaling is an important result to mention here too?

Yes, we will add a sentence to that end in the revised version of the manuscript: "*For all GCMs, downscaling with MAR significantly improves the representation of near-surface winds, except in the Transantarctic mountains and at the interface between the coast and the ocean.*"

**1. REFERENCES**

Davrinche, C., Orsi, A., Agosta, C., Amory, C., & Kittel, C. (2024). Understanding the drivers of near-surface winds in Adélie Land, East Antarctica. The Cryosphere, 18(5), 2239-2256.

---

## Author Response (AR1)

**POINT BY POINT RESPONSE TO REVIEWERS**

REVIEW OF DAVRINCHE ET AL., 2025 – FUTURE CHANGES IN ANTARCTIC NEAR-SURFACE WINDS: REGIONAL VARIABILITY AND KEY DRIVERS UNDER A HIGH-EMISSION SCENARIO

We thank the reviewers for their time and their valuable and helpful comments on the manuscript. We have implemented the following changes in a revised version.

Black = reviewer comment / Blue = author's comment / Italic = revised text.

**1. Response to reviewer 1**

The main points I raise in this review are regarding the application of the budget:

1. It's not quite clear from this analysis how the momentum budget decomposition was applied for each of the model configurations. Was your choice of parameters for diagnosing theta0 the same as in Davrinche et al. (2024)? Is there a way to test how robust your results are to the choice of Hmin?

For each model downscaled by MAR, we have performed the momentum budget decomposition following Davrinche et al. (2024). In this previous paper, in the supplement, we have validated our method and tested how robust our decomposition was to the choice of Hmin. We have also evaluated our pressure gradient force (PGF = KAT + LSC + THW) from the momentum budget decomposition (PGF) against the native pressure gradient force, which is a native output from MAR. It is also equivalent to comparing the native turbulence with the residual term. Because a lot of effort has been dedicated to the evaluation of the method in Davrinche et al., 2024, we are not planning on adding too much information about the evaluation in this paper.

However, as mentioned in Point 15, we have added a quantification of the error arising from closing the momentum budget. We have also updated the following sentence in the revised version:

L198: "The method is described extensively in Davrinche et al. (2024). For each model downscaled by MAR, we compute the momentum budget in the cross- and downslope directions and we decompose it into 6 different accelerations, defined as follows:"

2. The regional analysis is quite long and has a large figure count – you may be able to significantly improve readability by reducing the number of figure panels shown in Section 3.5, which focuses on a very specific question (the drivers of decreases). The vast majority of the changes shown in those regional panels are

already visible in Figure 5.

We thank the reviewer for this suggestion. We have taken it into account by moving figure 8 and 10 to the supplement.

3. A time-correlation analysis between the wind speed and budget terms would really help understand the role that surface forcing plays, instead of just looking at the change. How much variance in the July monthly mean between 1980 and 2100 do we explain with just VLSC from Equation 5, then how much when we add other terms?

We refer the reviewer to Section 4.4 of Davrinche et al., 2024 for a detailed study of the correlation between wind speed and the momentum budget terms. Given how long this paper already is, we did not want to repeat that analysis here. In this previous paper, we looked at the drivers of near-surface variability at the seasonal and 3-hourly timescale. Figure 1 and 2 (Figure 8 and 10 in Davrinche et al., 2024) are of specific interest to illustrate the role that surface plays in explaining the variance of July wind speed. We showed that the correlation between large-scale acceleration and total wind speed is high in locations where katabatic acceleration is weak. However, closer to the coast, none of the katabatic nor large-scale accelerations alone controls the variability at the 3-hourly timescale.

FIGURE 1. Seasonal cycle of 3-hourly nea-surface winds averaged over 10 years for (a) total wind speed, (b) wind speed equivalent to large-scale acceleration, (c) wind speed equivalent to thermal wind, (d) wind speed equivalent to advection, (e) wind speed equivalent to horizontal katabatic and (f) wind speed equivalent to turbulent accelerations. Note that the y-axis is different between panels a-d (|WS|,  $|V_{LSC}|$ ,  $|V_{THWD_{TD}}|$ ,  $|V_{ADVH}|$ ) and panels e-f ( $|V_{KAT}|$ ,  $|V_{TURB}|$ ).

FIGURE 2. (a) Average July 2010-2020 correlation coefficient of 3-hourly katabatic acceleration and wind speed (b) Average July 2010-2020 correlation coefficient of 3-hourly large-scale acceleration and wind speed (c) directional constancy of 3-hourly large-scale wind speed. (d, e, f): Mean of 3-hourly July 2010-2020 scalar product normalised by the norm of wind speed of (d) 3-hourly katabatic wind speed and total wind speed, (e) 3-hourly large-scale and total wind speed, (f) 3-hourly thermal-wind and total wind speed, (g) 3-hourly advection and total wind speed. For the 7 panels, the dotted black line corresponds to the line for which the correlation coefficient of katabatic acceleration and total wind speed reaches 0.5. Seven zones of higher correlations are indicated: (I), (II), (III), (IV), (V), (VI) and (VII)

On longer timescales, on the continent, the large scale forcing dominates interannual variability (Fig. 3), while changes in near-surface forcing result in lower frequency variability. Figure 3 also shows that the dominant drivers identified on the 1980-2000 period are still the same over 2080-2100. Here, we wanted to add to the previous paper by looking specifically at the change in the different terms between the two time periods, rather than to directly look at the 20 years time correlations.

We are also limited by the fact that the momentum budget decomposition is computationally heavy, and we could not run it for 100 years, but for 2 times 20 years. We are in essence comparing two time slices, and cannot do a statistical analysis on just two numbers, showing the difference, as we did, is more appropriate.

As we understand that we cannot expect all readers to have thoroughly read Davrinche et al., 2024, we have added some key results in the revised version of the manuscript:

140: "At present day, large-scale forcing dominates the variability of near-surface wind speed in the interior, while closer to the coast, none of the katabatic, nor large-scale accelerations alone controls the

3-hourly timescale variability (Davrinche et al., 2024). In future projections, [...]"

FIGURE 3. Proportion of variance of continental monthly July wind speed explained by the different accelerations for July 1980-2000 (black) and July 2080-2100 (green).

**Specific comments**

4. L30: A "thermal wind that acts to replenish the pressure low created by the downslope displacement of air". Could you describe what you mean by this in more detail – a thermal wind operates at large scales in a baroclinic atmosphere.

Do you mean the local thermal wind acceleration term or the thermal wind relationship used to calculate lsc?

We meant the thermal wind induced by horizontal gradients in the depth of the temperature deficit layer. It is explained in more details in Sec. 2.3.2. It quantifies the effect of baroclinicity at low levels. As we understand that it might confuse the reader in the introduction, we have rephrased it:

L36: "In addition, surface forcing creates two additional pressure gradients. The first is a katabatic pressure gradient, which is proportional to the strength of the temperature inversion and the slope angle. The second is a local thermal wind pressure gradient, which is created by horizontal gradients in the depth of the temperature deficit layer. Thermal wind acts to replenish the pressure low created by the downslope displacement of air."

5. Intro: please briefly review Bintanja et al. (2014) and consider the advancements made here relative to that research. I see you reference them later – it's worth signposting early on.

The reference is now mentioned earlier in the introduction:

- L27: Near-surface Antarctic winds result from both large-scale and surface pressure gradients (Van den Broeke and van Lipzig, 2002; Bintanja et al., 2014a; Davrinche et al., 2024)
- L43: On the one hand, the increase in GHG concentration causes a decrease in outgoing longwave radiation (Mitchell, 1989). As a consequence, the temperature inversion and thus the katabatic forcing should decrease (Van den Broeke and van Lipzig, 2002; Bintanja et al., 2014b).
- L75: "In addition to Bintanja et al. (2014b), we evaluate the representativeness of the results by performing this analysis on four recent CMIP6 GCMs carefully selected on their ability to represent the large-scale circulation in polar regions. It enables us to mitigate single-model analysis issues and to test how robust potential changes are."
- 6. L54: I'm not sure if dynamical downscaling alone ensures a physically realistic simulation of boundary-layer dynamics. Rephrase perhaps?

Yes, it is not just the downscaling, but also the better model physics over snow in MAR that leads to improvements. We have rephrased it in the revised version: L66: "This ensures a better resolution of the ice sheet topography as well as a more realistic simulation of boundary layer dynamics achieved through adapted parametrizations of the interactions between the snow/ice surface and the atmosphere, as well as higher resolution vertical spacing near the surface."

7. L100: Why select July and not the more usual climatological season of JJA?

The cost of computation and storage of the momentum budget terms for the four downscaled models is high and we could not afford saving all variables. Hence

the limited number of studied months. We have added the following sentence: L108: "For computational cost purposes, our study focuses on the winter month of July."

8. The supplement is very large and there's a lot of flipping back and forth between the main text and the supplement. If there is a way you can reduce the size of the supplement it would improve the flow. S1 and S2 are one equation I believe?

We feel that removing parts of the supplement would weaken our analysis. However, in order to improve the flow, we have removed as many back and forth flipping as possible. For instance:

- L390: As Ross ice shelf is now presented in the supplement, this has removed a supplement-main body flip
- L418: As Shackleton ice shelf is now presented in the supplement, this has removed a supplement-main body flip
- 9. S1.1: is the relative uncertainty the standard error?

Not exactly, the relative uncertainty is the standard error divided by the mean value:

Relative uncertainty =
$$\frac{\frac{\sigma_{July}}{\sqrt{N_{July}}}}{|V_{July}|}$$
 (1.1)

We acknowledge that it might be complicated for the reader, and have updated this sentence in the revised version of the manuscript:

1109: "In order to test whether datasets are long enough to be representative of a climatological period, we compute using ERA5 the minimum value of  $N_{July}$  for which the standard error on the mean value of the July wind speed between 1980 and 2020 is inferior to 5% of the mean value (see Supplementary Section S1.1)"

10. L124: why calculate the metrics for December and the annual mean if we are only focused on July?

We wanted to give a more general result regarding the added value of downscaling by MAR regarding the representation of near-surface winds. We understand that it is not the primary focus of this paper, and have removed it. Therefore we present only July in the main body:

- We have computed the TPS for July only, and have moved the last 6 colums of Table 4 to the supplement.
- We have updated Fig. S3 as follows:

FIGURE 4. Score of the 28 pre-selected AWS stations compared to ERA5 for all July available AWS data Three metrics are considered: the correlation coefficient (R), the normalized mean bias (B) and the normalized standard deviation ( $\sigma_N$ ). Each metrics for each station gives a score equal to -1 and 1 depending on its performance (see Sec. 2.1.4). Positive values indicate a good performance. (a) Scores for each metrics and for each stations. (b) Sum of all individual scores. Red solid line on the colorbar indicates the threshold under which stations are excluded based on their comparison with ERA5. Those stations are shaded in blue.

- Have updated underlined stations on Fig. 1
- Have updated Table 2
- Have updated Fig. 3 as follows:
- We have updated Sec. 3.1
- We have removed Fig. S4
- 11. Supplement L16: check the reference to 'Figure 5 in the manuscript'.

It should indeed refer to Figure 3, thit has been updated in the revised version of the manuscript.

12. Supplement: in my PDF Figure S3 shows after Figure S4.

It has been corrected in the revised version of the manuscript.

13. Figure S3: I am a bit confused by the (b) panel colour bars. What is the left and right coloured bar showing? Maybe it would be simpler to show each

FIGURE 5. (a) Altitude of the selected stations. Mean normalized bias (B) for wind speed with regard to the AntAWS observations (B =  $(|\overrightarrow{V_{GCM}}| - |\overrightarrow{V_{AntAWS}}|)/|\overrightarrow{V_{AntAWS}}|$  for (a) and B =  $(|\overrightarrow{V_{MAR-GCM}}| - |\overrightarrow{V_{AntAWS}}|)/|\overrightarrow{V_{AntAWS}}|$  for (b)) for the 24 selected AntAWS stations, computed for July (b) using the GCMs, (c) using the GCMs downscaled by MAR.

individual TPS on the grid and e.g. hatch the gridcells which pass the threshold?

As mentioned above, we have updated Figure S3 and replaced the heatmap by a regular map with blue shaded stations indicating stations that do not pass the threshold.

14. L203 Is this strictly speaking the boundary layer? You imply here that the height at which the vertically integrated temperature deficit becomes zero is the top of the boundary layer. In East Antarctica however the temperature deficit can extend to 4km height (see e.g. Figure 3 in van den Broeke and van Lipzig, 2003). This is much deeper than the top of the stable boundary layer, which vdB and vL say is 'poorly constrained'. Is it not more correct to say that it's just the vertical integral of the temperature deficit? My understanding is that the temperature deficit can extend far above the boundary layer, which over the plateau may be e.g. 10-150m at Dome C, Pietroni et al., 2012: https://doi.org/10.1007/s10546-011-9675-4

Yes, it is correct. It is now mentioned in the revised version of the manuscript: L211: " $\hat{\theta}$  is the vertically integrated potential temperature deficit from the top of the inversion layer. Above the inversion layer, as  $\theta = \theta_0$ , both  $\Delta \theta$  and  $\hat{\theta}$  become

zero."

15. L226 does the residual here also encompass any errors from closing the budget (e.g. finite difference approximations) or is it directly output from the model?

Yes, we have detailed that L237: "The residual term (TURB) encompasses vertical advection (which is weak), turbulent drag (which opposes the other accelerations and is strong when the wind speed is high) and potential errors arising from closing the momentum budget. A comparison of MAR's native turbulent acceleration and our recomputed residual turbulence as detailed in Davrinche et al., 2024 enables us to conclude that the error resulting from closing the budget in July is weak compared to the absolute value of the turbulence (ie ~ 10 % for all models)."

16. Figure 3a – what is the x-axis here

The former x-axis represented the stations' number. We removed these values to only keep the ticks (see Fig. 5).

17. Section 3.1 no need to restate this first para, or move to the introduction.

This paragraph has been moved to the introduction in the revised version.

18. L270 figure panel reference needed

The panel reference have been added in the revised version.

19. Figure 3: I think I missed what the collocation method is? Are you using nearest neighbour or bilinear? Is MAR regridded to the same grid as the ESMs? If not it would be useful to do this as an additional analysis to just check if the added value comes from being able to collocate a gridpoint closer to the location of the AWS in MAR.

We forgot to mention it, but we did regrid the GCMs on MAR's grid using a bilinear interpolation. We have added it in the revised version: L160: "They are regridded using a bilinear interpolation on MAR's grid."

20. Figure 4: (v) not quite able to tell but it looks like this is not the Ronne ice shelf? It may be worth checking – in my understanding the Ronne hugs the peninsula and the Filchner ice shelf is east of that.

Yes, according to this detailed map (https://images.nationalgeographic.org), region (V) is closer to Filchner ice shelf than to Ronne ice shelf. This has been modified in the figures and in the main body of the revised manuscript.

21. Table 3: in my PDF this appears below Figure 4 (but referred to beforehand).

It has been corrected in the revised version.

22. L357: my understanding is this ^ is the vertical integral of the deficit rather than the depth of the layer

Yes, it has been corrected:

L354: "Associated with changes in  $\Delta\theta$ , the depth of the temperature deficit layer  $\hat{\theta}$  also changes. It reduces considerably on the continent, near the coastline (Figure S6), causing a reduction in thermal wind (Figure 5d)."

23. L357: please specify where in Figure S7 you are referring to for the coastline

As stated in the Point 22, we have added the group of words "on the continent", as we are focusing on onshore winds.

24. L359: in some regions (e.g. offshore of Adelie land) the thermal wind is a positive forcing term and does not oppose the katabatic wind so it doesn't necessarily increase wind speed if you reduce it.

We were focusing on onshore winds but forgot to mention it in this sentence. As stated in the points 22 and 23, we have added the group of words "on the continent".

25. L398: where are these regions where 'surface forcing can also contribute to significant wind speed increase'?

In L396, it is specified for "Ross ice shelf", but we should have mentioned the MAR-CNRM model. In this simulation, the large-scale acceleration decreases, and the observed significant increase in total wind speed can only be linked to changes in the surface forcing. We have changed the following sentence accordingly:

"L396: Although it is clear from the analysis of Adélie and Enderby Land that significant increases in the large-scale forcing drive changes in the near-surface wind speed, the analysis of Ross ice shelf (Figure S8, MAR-CNRM) indicates that surface forcing can also contribute to a significant increase in wind speed."

26. L455 you imply here that some regions have an increased wind speed due to surface forcing, and it's true that the surface forcing does increase (kat+thw) in some regions but I don't see these mapping onto obvious increases in wind speed.

What we meant here was that wind speed resulting ONLY from surface forcing was overall increasing. L448: "Because the thermal wind opposes the dominant direction of the downslope winds in the sloped regions of Antarctica  $\sim 250 \mathrm{km}$  from the coastline (Davrinche et al., 2024), a weakening of the thermal wind forcing

increases the resulting wind speed and compensates for the decrease in the katabatic acceleration in these onshore regions. The compensating effect of thermal wind is particularly pronounced in coastal East Antarctica where it often exceeds the decrease in katabatic forcing, leading to an overall increase in the wind speed resulting from the surface forcing only."

28. PIG -> Amundsen embayment region?

This has been updated in the revised version.

29. L454 regional specifics would be helpful here as this compensating effect only applies where the katabatic winds are active

Yes, it is true. As we do not want to introduce new categories of regions depending on the elevation or slope (as we did in (Davrinche et al., 2024)), we have added some descriptions in the sentences instead: "L448: Because the thermal wind opposes the dominant direction of the downslope winds in the sloped regions of Antarctica ~ 250 km from the coastline (Davrinche et al., 2024), a weakening of the thermal wind forcing increases the resulting wind speed and compensates for the decrease in katabatic acceleration in these onshore coastal regions. The compensating effect of thermal wind is particularly pronounced in coastal East Antarctica where it often surpasses the decrease in katabatic forcing, leading to an overall increase of the wind speed due the surface forcing only."

30. Section 4: I may have missed it but I think the added value of dynamical downscaling is an important result to mention here too?

Yes, we have added a sentence to that end in the revised version of the manuscript:

L426: "For all GCMs, downscaling with MAR significantly improves the representation of near-surface winds, except in the Transantarctic mountains and at the interface between the coast and the ocean."

**2. Response to reviewer 2**

Major comments:

1. Many parts of the manuscript come across as rather unpolished and the writing disjointed. This really needs to be improved.

We have taken this comment into account, and have polished the revised version of the manuscript.

2. For example, many of the sentences in the Introduction claim something but do not include a citation for evidence. So sentences such as 'On the one hand,

the greenhouse warming causes an increase of the incoming longwave radiation.'

We have replaced this sentence as follows:

L 43: "On the one hand, the increase in GHG concentration causes a decrease in net upward longwave radiation at the surface (Mitchell, 1989). As a consequence, the temperature inversion and thus the katabatic forcing should decrease (Van den Broeke and van Lipzig, 2002; Bintanja et al., 2014b)."

In general, we have rewriten large parts of the introduction with careful attention to the inclusion of citations for any idea described.

3. and 'Although there is a consensus on the reduction of surface forcing in climate projections'.

We have added the following references L55 in the revised version: "(Van den Broeke and van Lipziq, 2002; Bintanja et al., 2014b)."

4. There are also incidences of repetition, such as in the Introduction with something along the lines of 'which is proportionate to the strength of the temperature inversion' mentioned twice

We have spotted a repetition line 34 and have removed it in the revised version.

5. and in the methods and Introduction which both mention something along the lines of 'Because of their resolution, GCMs are not expected to perform well in locations with complex topography.'

We did not mention specifically the resolution of GCMs in the introduction, but rather their ability to take into account complex topography, land—sea contrasts, boundary-layer and convective processe (L51).

6. Other instances are the preambles/motivation before the results, which just say in a slightly different fashion what was said before. Please remove all repetition, and remember that your audience/readers only need to be told something once.

We were following the article writing guidelines developed in Plaxco, 2010: "The first sentence of each paragraph should tell the reader what you expect them to get out of the paragraph that follows, which makes their job of following it far easier. Put another way; use the opening sentence of your paragraph to state your argument, and the rest of the paragraph to make your argument." However, we have taken into account your comment and made our best to remove all opening sentences.

7. Also there are typos, such as '(e.g. north of Ross and Amery ice shelves and north of the Peninsula' in the Introduction (so no closure of parentheses).

We apologize for this mistake, it has been corrected in the revised version.

8. Mistakes such as AWS defined, and the phrase automatic weather station still used.

We have spotted two AWS definitions: l66 in the preamble and l 71. Because we have removed the preamble in the revised version, there is no longer a repetition.

9. Very random / ad hoc approaches such as using m/s in one sentence and km/hr in the following sentence (methods). These give the feel of a rushed writing process, and of a manuscript submitted before it was really ready.

We have spotted this line 84, and modified it in the revised version of the manuscript.

10. There are also parts of it which are disorgansied, such in section 2.1 mentioning ERA5, and then ERA5 not being explained until later (also it's not explained in a logical fashion from the methods that ERA5 is being used to select the GCMs.).

In Sec 2.1, ERA5 was mentioned but not explained in the preamble. Because we are removing all preambles in the revised version, ERA5 is no longer mentioned before being explained.

11. Poor paragraph structure such as section 2.1.2.

We have updated this paragraph: "ERA5 is the latest reanalysis produced by the European Centre for Medium-Range Weather Forecasts (Hersbach et al., 2020). Its horizontal spatial resolution is ~ 31 km and outputs are given at a hourly frequency. The assimilation system (IFS Cycle 41r2 4D-Var) uses 10 members to produce a 4D-Var ensemble of data assimilation (Hennermann and Guillory, 2019). Among various reanalysis products (MERRA-2, JRA-55, ERAI, NCEP2, and CFSR), ERA5 has been shown to perform best in capturing monthly averaged wind speeds (Dong et al., 2020)."

12. Finally, some odd sentences such as 'We focus on the Antarctic continent, which is the source region of the katabatic forcing' in the final paragraph of the Introduction.

This has been changed in the revised version:

L70: "We focus on the Antarctic continent, where katabatic winds are developing in the sloped regions due to the quasi permanent radiative cooling by the ice sheet (Phillpot and Zillman, 1970), and on the winter season, as it is the season for which both the katabatic forcing and the mean wind

speed are the highest (Davrinche et al., 2024)."

13. Methods: Out of the blue it is mentioned that the subset of AWSs are selected based on their ability to represent ERA5. This is not justified. Additionally, this seems a rather strange choice, as ERA5 would also struggle to represent steep coastal gradients, so also do poorly representing katabatic winds. So justification is clearly required.

The subset of AWS is not selected based on their ability to represent ERA5. On the contrary, we show that ERA5 is not able to reproduce correctly surface wind speed in some locations with complex topography. As we do not expect GCMs to perform better than the reanalysis over the period of available AWS observations (as stated L129), we have decided to exclude AWS that were already misrepresented in ERA5, which assimilates observations in Antarctica. In the end, we exclude stations located in the Transantarctic mountains and at the interface between the continent and the ocean, which follows expectations. However, we wanted to use a rigorous method to exclude those stations.

We understand that some sentences in the manuscript might suggest that we exclude stations based on their ability to represent ERA5 instead of based on the ability of ERA5 to represent the climatology of the AWS. The following changes have thus been implemented in the revised version:

- L123: The title "2.1.4 Exclusion of sites near complex topography based on performance of ERA5" has been changed to "2.1.4 Exclusion of sites near complex topography"
- L138: "These four stations exhibit the largest biases in terms of temporal variability (R < 0.3 and σN > 2, which indicates that the variability in ERA5 is underestimated) and mean amplitude (B > 30%, which indicates that ERA5 overestimates the mean value of the wind speed). Additionally, these stations are all located at the foot of the Transantarctic mountains (Fig. 1), which justifies their exclusion in the quantitative analysis."
- 13. The correction to the AWS dataset is also poorly explained (Equations 1 and 2) its not even clear what is being corrected, and what 1-3 and 1-6 refers to.

We have explained more clearly what equations 1 and 2 refer to:

L86: "According to the logarithmic theoretical profile of wind speed in the boundary layer, with a constant roughness length  $z_0 = 1$  mm (Vignon et al., 2017), we estimate the maximum correction between wind speed measured at the real height of the sensor and wind speed at 3m to be between -10 % (for the correction from 1 to 3m) and 7% (for the correction from 6 to 3m) of the theoretical

value:

$$correction_{6m-3m} = \frac{log(\frac{6}{z_0})}{log(\frac{3}{z_0})} = 1.07$$
(2.1)

$$correction_{1m-3m} = \frac{log(\frac{1}{z_0})}{log(\frac{3}{z_0})} = 0.90$$

$$(2.2)$$

"

14. Selection criteria for GCMs: This seems to state that their performance in the Arctic is also taken into account, which is completely unjustified.

For practical reasons, we did not want to use the entire CMIP6 range of models in our study. We aimed to use a small subset of 4-5 models that 1) are not too wrong, and 2) represent a range of possible climate projections. Among CMIP6 models, we have selected those that performed the best in the Antarctic. As both poles share common physical processes, we have selected among models that performed well in the Antarctic those who also performed well in the Arctic to get to a reasonable number of models. Additionally, the choice of these four models for our study is supported by another study by Williams et al. (2024) where these models were classified among the best performing ones in Antarctica, in winter when comparing their sea ice extent, surface air temperature, zonal wind at 850 and 50 hPa to ERA5.

15. There are a lot of locations mentioned, but I don't think they are always shown on a map.

All stations are shown on Fig. 1. Regions of interest are shown on Fig.4. We have added the Transantarctic mountains and the Plateau on Fig. 1 in the revised version.

16. Minor comments (this is just a selection as there are a lot of 'minor' concerns that need to be addressed by a very thorough revision of the paper)

We have paid extra attention to re-reading and correcting all typos in the manuscript.

17. Abstract: Not clear what the distinction between katabatic and thermally driven winds is. I think some explanation of the term 'thermally driven' is necessary here, as otherwise the reader is lost.

We have added some explanations in the abstract:

L11: "These drivers include local forcings related to the net radiative cooling by the iced surface as well as large-scale forcing. We distinguish two types of local forcing: katabatic forcing (linked to the presence of a slope)

**and thermal wind forcing, which arises from horizontal gradients in the depth of the radiatively cooled surface layer."**

Additionally, we have changed the sentence in the introduction to: L36: "In addition, surface forcing creates two additional pressure gradients. The first is a katabatic pressure gradient, which is proportional to the strength of the temperature inversion and the slope angle. The second is a local thermal wind pressure gradient, which is created by horizontal gradients in the depth of the temperature deficit layer. Thermal wind acts to replenish the pressure low created by the downslope displacement of air."

**18. The introduction mentions one mode of variability, the SAM. But what about the Amundsen Sea Low?**

We have mentioned SAM as it is the dominant mode of variability in the southern hemisphere (Thompson and Salomon, 2002). However, we acknowledge that we should also have mentioned the influence of ENSO. We have added it in the revised version of the paper:

L29: "Large-scale forcing is intrinsically linked to the leading modes of variability in the Southern Hemisphere: the Southern Annular Mode (SAM) and the El Niño-Southern Oscillation (ENSO). The SAM is quantified by the SAM index, which represents the zonally averaged sea-level pressure gradient between 40°S and 65°S (Marshall, 2003). ENSO is characterized by the Southern Oscillation Index (SOI), computed as the sea-level pressure difference between Tahiti and Darwin (Bromwich et al., 2004). Both SAM and ENSO influence the strength and position of the Amundsen Sea Low, a persistent low-pressure center in the Amundsen Sea sector (Raphael et al., 2016), which in turn modulates the frequency and trajectories of cyclones in West Antarctica (Fogt et al., 2012)."

L45: "On the other hand, the increase in GHG concentration drives the SAM towards a more positive phase by the end of the 21st century (Miller et al., 2006; Fogt and Marshall, 2020; Goyal et al., 2021) while the effect on the SOI remains highly uncertain (Beobide-Arsuaga et al., 2021; Ren and Liu, 2025)."

**19. The katabatic winds are also dependent on the size of the slope.**

Yes, we did not mention it in this sentence, because we wanted to highlight the dependence of the katabatic acceleration to the temperature inversion, as the slope does not change between the two time periods, but we agree that it might be misleading for the reader, and have added it in the revised version: L36: "In addition, surface forcing creates two additional pressure gradients. The first is a katabatic pressure gradient, which is proportional to the strength of the temperature inversion and the slope angle. The second is a local thermal wind pressure gradient, which is created by horizontal gradients in the depth of the temperature deficit layer. Thermal wind acts to replenish the pressure low created by the downslope displacement of air."

20. Section 2.1.3: Not sure why the comment on the length of observations in the summer season is necessary.

Yes, indeed, it was a mistake, we were referring to **austral winters**. This has been corrected in the revised version.

21. And shouldn't the number of AntAWS stations mentioned here, actually be mentioned in section 2.1.1. Comes across as disorganised.

The minimum number of observations is justified based on the variability of near-surface winds derived from ERA5. Consequently, this discussion cannot be included in Section 2.1.1, as it precedes the introduction and description of ERA5. However, we have added the total number of AWS in Section 2.1.1:

L 83: "For all 267 stations (except Zhongshan) [...]"

22. Section 2.1.4: At least the third time that GCMs issues over representing complex orography has been mentioned. Repetition. Makes the manuscript look extremely disorganised and amateurish.

We have removed a repetition by discarding the preamble. The inability of GCMs to represent complex topography is now only mentioned once.

23. Line 118. Typo. Grill -¿ Grid

Yes, this has been corrected in the revised version.

24. Methods: Its not clear what the term 'Implausibility' is being used here for.

We understand this comment and have given more details in the revised version:

L166: "Fraction of implausibility" is defined for each metric as the portion of the surface where the difference between historical averages in the model and ERA5 is greater than a plausible threshold set at 3 times the ERA5 interannual standard deviation (Agosta et al., 2022)."

25. Section 2.3.2: Poor paragraph structure.

We have modified the structure of this paragraph by moving the comment on the active terms L229, after the description of the pressure gradient.

26. Section 3.1: The preamble here is inappropriate / repetition. This material should be in the Introduction or Methods, not repeated at the beginning of the results section. This weakens the paper and makes it look disorganised.

The preamble here has been removed in the revised version. Parts of the preamble have been moved to L49 in the introduction.

27. Section 3.3: Similar comment to above, no need for the preamble prevalence

The preamble has been removedn.

28. Line 319: SAM defined again

Yes, we have removed the bracket.

29. Section 3.3.1: Huge amount of repetition on how SAM will change.

We have thoroughly rewriten the section to avoid unnecessary repetitions. However, it is also a matter of style and efficiency to have a few repetitions of key concept in a 25 pages manuscript. The SAM trends are described in 1 sentence in the introduction L45: "The increase in GHG concentration drives the SAM towards a more positive phase by the end of the 21st century (Miller et al., 2006; Fogt and Marshall, 2020; Goyal et al., 2021)" and then once in the results:

L319: This result is in agreement with previous studies that showed that the already observed increasing positive trend of the SAM will likely continue in response to increasing greenhouse gases and after the recovery of the ozone hole (which offsets the strengthening of the SAM (Bracegirdle et al., 2008))."

We checked for repetitions in the paper and have removed most of them. But we we would like to keep the sentences on how SAM will change as we consider that 2 mentions, 16 pages apart, does not harm the flow of the paper.

**3. REFERENCES**

Davrinche, C., Orsi, A., Agosta, C., Amory, C., & Kittel, C. (2024). Understanding the drivers of near-surface winds in Adélie Land, East Antarctica. The Cryosphere, 18(5), 2239-2256.

---

## Author Response (AR2)

**POINT BY POINT RESPONSE TO REVIEWERS**

REVIEW OF DAVRINCHE ET AL., 2025 – FUTURE CHANGES IN ANTARCTIC NEAR-SURFACE WINDS: REGIONAL VARIABILITY AND KEY DRIVERS UNDER A HIGH-EMISSION SCENARIO

We thank the reviewers for their time and their valuable and helpful comments on the manuscript. We have implemented the following changes in a revised version.

Black = reviewer comment / Blue = author's comment / Italic = revised text.

**1. Response to reviewer 1**

1. L161: should Gill be underlined in Figure 1? Yes that is correct. We had previously only underlined the stations on the right panel, but we have also added it on the left one in the revised version.

2. L370: unclear if this is causal or just coincident

We meant it in a more coincident way, since there is no clear consensus in the literature regarding the effect of sea-ice retreat on the mid-latitude jet position nor strength. Kidston et al., 2011 suggest that any future decrease in Antarctic sea ice are unlikely to have a profound effect on the Southern Hemisphere mid-latitude circulation while Bader et al., 2012 findsthat reduction of SH sea ice leads to an equatorward shift of the mid-latitude jet. Due to the complex interplay and feedback between temperature changes, sea-ice loss, oceanic heat transport and jet modifications, it is complex to say whether sea ice loss can have a direct impact on the changes in the jet position. In order to make it clearer for the reader that we are not implying any link of causality, we will rephrase the sentence as follows:

L320: The pattern of increase in westerlies coincidentally appears to follow closely changes in the extent of sea ice, shown in thick black lines in Figure 4.

**3. L492: 21st**

This has been modified in the revised version

4. L501: "the decrease in coastal easterlies in all models is stronger in the MAR downscaling, where changes in the surface forcing are likely better represented" – the paper focuses mostly on scalar wind speed (sfcWind), without an analysis of wind components. It could be worth rephrasing where you use 'easterlies' in this paper for clarity? An increase in sfcWind could mean weakening easterlies (replaced by westerlies) or strengthened easterlies and vice versa

Yes, we understand this comment as we have indeed not shown any map of the

FIGURE 1. Projection of changes in 10-m wind speed between 2080-2100 and 1980-2000 for (a) MAR-MPI, (b) MAR-CNRM, (c) MAR-UKESM and (d)MAR-IPSL. Superimposed are the average wind vector for July 1980-2000 (black arrows) and July 2080-2100 (red arrows).

changes in wind direction. However, projected changes in mean wind direction or minor, as shown on Fig. 1. Therefore, we will change the following sentence, where me mention the weakening easterlies for the first time:

L329: Everywhere else in Antarctica, MAR-IPSL and MAR-MPI project an overall increase in large-scale acceleration, while MAR-UKESM and MAR-CNRM exhibit some significant weakening of coastal easterlies (with minor changes in the mean wind direction) on Shackleton ice shelf and in Queen Maud Land.

**2. Response to reviewer 2**

1. The second paragraph of the Introduction is not a stand-alone paragraph – it's a single sentence, and consequently makes the Intro seem disjointed.

**The paragraph break has been removed.**

2. The paragraphs from lines #229 to #233 are also not suitable as stand-alone paragraphs.

Both paragraph breaks has been removed.

3. I don't think that a paragraph break is necessary in Line #295.

The paragraph break has been removed.

4. The remark in the final paragraph of the Introduction of the causes of katabatic winds should be mentioned much earlier, when katabatic winds are first introduced.

This remark has been moved to line 36/37

5. The use of 'Bedmachine' DEM requires a reference

We have added a reference to Morlighem et al., 2020.

6. line #160 gives short names for the GCMs and says there are referred to these hereafter, but the long names are then used in Table 2 and Line #169

Yes, it is true. Therefore we have introduced the short names line 169, at the end of the section instead.

7. Sentences such as 'They are regridded using a bilinear interpolation on MAR's grid.' are rather careless, as its not clear what is being regridded from this sentence (Output from these models are regridded ...).

We have modified the sentence as suggested:

L160: Output of these models are regridded to MAR's 35km polar stereographic grid using a bilinear interpolation.

8. line #174, which starts by mentioning storylines, then jumps to climate sensitivity, and then back to explaining the storylines. The justification for mentioning the different future changes in sea ice extent or stratospheric polar vortex is not very clear.

We understand this comment. Therefore we have:

- Moved the following sentence "Note that all of these models are Earth System Models, except for CNRM-CM6-1 which does not include interactive ocean biogeochemistry nor atmospheric chemistry (Voldoire et al., 2019)" right before the first mention of storylines, so that it does not interrupt the sentences about the storylines.
- We understand that the explanation for mentioning the ECS, SIE and SPV was not very clear and rephrased it such as follows:

The choice of these four models for our study is supported by another study by Williams et al., 2024 where these models were classified among the best performing in winter when comparing their sea ice extent (SIE), surface air temperature, zonal wind at 850 and 50 hPa to ERA5. Furthermore, these models are representative of the large variability of plausible patterns of responses to climate change among CMIP6 models and can be expected to exhibit different patterns in wind-speed changes by the end of the 21st century. For example, Williams et al., 2024 noted that they correspond to different storylines for Antarctica, using winter SIE and Stratospheric Polar Vortex (SPV, linked to the strength and position of the surface westerlies, Table 2) as predictors. Additionally, they have different Earth's Equilibrium Climate Sensitivity (ECS, corresponding to the change in temperature at equilibrium that would result from a doubling of  $CO_2$ ), which is a proxy for the intensity with which the model warms the Earth's surface temperature. While UKESM has one of the strongest ECS of all CMIP6 models, MPI exhibits one of the lowest.

9. Line #429: XXIst century? Is this the 21st century? I have never seen this written like this before.

Yes, it is 21st century in roman numerals. It has been changed in the manuscript.

10. I simply don't see much of a Discussion of the results here. Such as putting them into context with the current scientific understanding. Properly explaining them. Referencing other similar work / results / studies. Etc. This section actually only mentions three papers, one of which the authors led. And there is again poor paragraph structure (line #462). This entire section rather comes across as a series of statements about the results, so much more Conclusion than Discussion. This section needs to be strengthened considerably, with a lot more thought put into it.

We have reorganised and rewritten the entire discussion. Major changes are the following ones

- We have separated the section into 1. Discussion and 2. Conclusions
- The conclusion section incorporates the majority of the sentences from the initial conclusion but has been reorganised and care has been given to the structure of the paragraphs
- The discussion section is completely new and incorporates more comparison with results from 7 previous studies

**3. REFERENCES**

Bader, J., Flügge, M., Kvamstø, N. G., Mesquita, M. D., & Voigt, A. (2013). Atmospheric winter response to a projected future Antarctic sea-ice reduction: A dynamical analysis. Climate Dynamics, 40(11), 2707-2718.results

Kidston, J., Taschetto, A. S., Thompson, D. W. J., & England, M. H. (2011). The influence of Southern Hemisphere sea-ice extent on the latitude of the midlatitude jet stream. Geophysical Research Letters, 38(15).

Morlighem, M., Rignot, E., Binder, T., Blankenship, D., Drews, R., Eagles, G., ... & Young, D. A. (2020). Deep glacial troughs and stabilizing ridges unveiled beneath the margins of the Antarctic ice sheet. Nature geoscience, 13(2), 132-137.